# Divisiveness-Consistent Label Distribution Learning

Yunan Lu [1]  Haitao Wu [2]  Weiwei Li [3]  Lei Yang [1]  Xiuyi Jia [2]

## Abstract

Label Distribution Learning (LDL) is an effective learning paradigm for predicting entire conditional label distributions, improving the trustworthiness of predictions in risk-sensitive tasks. Although previous LDL methods achieve satisfactory performance on conventional evaluation metrics, they generally overlook the divisiveness within label distributions, i.e., the propensity of label distribution to exhibit dissension between semantically opposing labels, which is an essential indicator of the practical decision risk. Therefore, we propose a divisiveness-consistent label distribution learning framework to quantify and preserve the divisiveness information. First, we formalize a divisiveness measure that satisfies the axiomatic property of polarity monotonicity to quantify the divisiveness information. Second, we theoretically demonstrate the inconsistency between conventional loss functions and divisiveness error. Besides, in order to address the adversarial gradient problem arising from directly minimizing the divisiveness error, we propose a pairwise divisiveness loss as an unbiased estimator of the original divisiveness error. Experiments confirm the effectiveness of the proposed method.

## 1. Introduction

The accurate estimation of the entire conditional distribution of labels (commonly referred to as the label distribution) given a set of feature variables, beyond the point estimates such as the mean or mode, has garnered growing attention in both statistics and machine learning (Hothorn et al., 2013; Gneiting & Katzfuss, 2014; Thielmann et al., 2024).

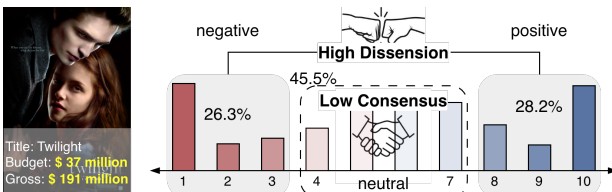

(a) High divisiveness: 26.3% opponents vs. 28.2% supporters.

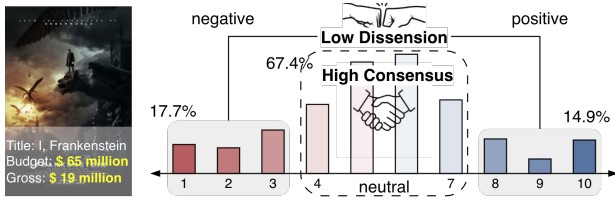

(b) Low divisiveness: 17.7% opponents vs. 14.9% supporters.

*Figure 1.* Intuitive motivation behind the concept of divisiveness.

This focus stems from the critical need for complete distribution information in the applications that are sensitive to risk, extreme values, or uncertainty. In pursuit of this goal, researchers have developed various methods, such as model calibration (Lichtenstein et al., 1977; Vaicenavicius et al., 2019; Song et al., 2019) and mixture density networks. These methods aim to predict the true label distributions from instances only with aggregate statistics (e.g., the mean or mode), particularly valuable in the practical settings where true label distributions are hardly available.

Nevertheless, a large number of real-world scenarios (Lee et al., 2021; Nguyen et al., 2012; Yang et al., 2014; 2015; Zhao et al., 2023; Liu et al., 2014) provide direct access to the ground-truth label distributions. Taking crowdsourced learning as an example, the labeling results for each instance can be normalized as the proportion of participants who assign each label to the corresponding instance. The problem of learning from the instances with label distributions is termed Label Distribution Learning (LDL) (Geng, 2016). Compared to the learning tasks only with aggregate statistics, LDL is capable of predicting the label distributions of unseen instances more accurately, since LDL is directly supervised by the ground-truth label distributions.

Although previous studies on LDL have proposed numerous effective algorithms, they largely overlook a critical

[1]Department of Computing, The Hong Kong Polytechnic University, Hong Kong, China [2]School of Computer Science and Engineering, Nanjing University of Science and Technology, Nanjing, China [3]College of Computer Science and Technology, Nanjing University of Aeronautics and Astronautics, Nanjing, China. Correspondence to: Xiuyi Jia <jiaxy@njust.edu.cn>.

*Proceedings of the 43rd International Conference on Machine Learning*, Seoul, South Korea. PMLR 306, 2026. Copyright 2026 by the author(s).

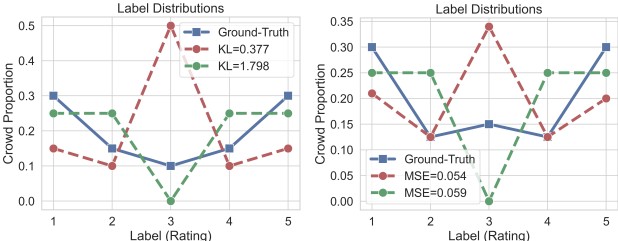

(a) Representative examples of the metric inconsistency.

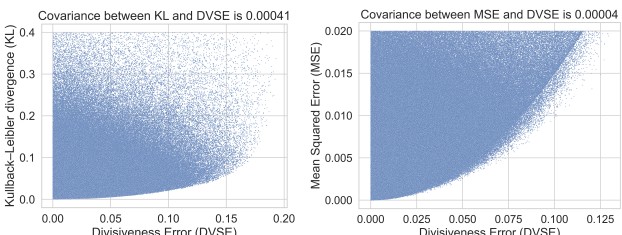

(b) Empirical joint distribution under near-convergence conditions.

*Figure 2.* Inconsistency between the conventional loss functions (including KL divergence and MSE) and divisiveness error.

characteristic underlying the label distribution: divisiveness, i.e., the propensity of label distribution to exhibit dissension between semantically opposing labels, which is an essential indicator of the potential risk of the practical decision making. As illustrated in Figure 1, in the rating prediction task where the ratings represent the crowd preference for a movie, a U-shaped rating distribution in Figure 1(a) (where ratings are concentrated at both high and low extremes) signals intense polarization of crowd preference, which entails greater commercial risk and opportunity than a bell-shaped distribution in Figure 1(b) (where ratings cluster around a median value) (Geng & Hou, 2015). In the context of urban governance, policies characterized by U-shaped public opinion distributions, reflecting a high degree of polarization, are prone to provoke widespread social controversy and significant implementation challenges.

However, the loss functions employed in most LDL methods, such as Kullback-Leibler (KL) divergence and Mean Squared Error (MSE), focus on the global distributional proximity. This objective is inherently inconsistent with the divisiveness characteristic, as these loss functions fail to distinguish between consensus and polarization within the predicted label distribution. On the one hand, Figure 2(a) provides a representative example of the inconsistency between conventional loss functions and divisiveness. From the perspective of global distributional proximity, the red label distribution prediction achieves lower KL divergence and MSE relative to the blue (ground-truth) label distribution compared to the green prediction. However, from the perspective of divisiveness, the green prediction, which shares the similar U-shaped profile of the ground-truth, ex-

hibits a consistent level of divisiveness. In contrast, the bell-shaped red prediction demonstrates substantially higher divisiveness error compared to the green prediction despite its better performance on global distributional proximity. On the other hand, the joint distribution visualized in Figure 2(b) reveals a critically low covariance between the conventional loss values and the divisiveness error, empirically demonstrating that the minimization of conventional loss functions is not equivalent to the minimization of divisiveness error.

Therefore, this paper proposes a divisiveness-consistent label distribution learning framework to explicitly quantify and preserve the inherent divisiveness within label distributions. Our research is structured around two core components: measurement of divisiveness and learning with divisiveness. In terms of measuring divisiveness, we formalize a divisiveness measure according to systematic case studies on the interplay between divisiveness and the morphology of label distributions. To establish theoretical justification for the proposed measure, we introduce an axiomatic property that any principled divisiveness measure should satisfy: polarity monotonicity, and prove that our defined measure preserves this monotonicity. Additionally, we critically examine alternative divisiveness measures and delineate their limitations relative to our formulation. In terms of learning with divisiveness, we theoretically quantify the inherent inconsistency between conventional loss functions and divisiveness error, thereby establishing the necessity of a dedicated divisiveness loss function. Subsequently, we identify that directly minimizing the absolute divisiveness error between the ground-truth and prediction induces adversarial gradients, which impedes stable convergence. Accordingly, we propose a pairwise divisiveness loss as a surrogate, which formally constitutes an unbiased estimator of the original divisiveness error. Finally, experiments demonstrate the effectiveness of our proposal. The main contributions are summarized as follows:

- **Principled Formalization of Divisiveness Measure**. We formalize a divisiveness measure satisfying the axiomatic property of polarity monotonicity, which accurately captures the propensity for dissension between semantically opposing labels and provides a theoretical foundation for the divisiveness-consistent learning.

- **Theoretical Analysis of Loss Inconsistency**. We theoretically analyze the inconsistency between conventional loss functions and divisiveness error, which reveals why optimizing global distributional proximity alone cannot preserve divisiveness information.

- **Unbiased Surrogate of Divisiveness Error**. We propose a pairwise divisiveness loss as an unbiased surrogate of divisiveness error, which ensures stable model convergence while effectively harmonizing global distributional proximity and divisiveness error.

## 2. Related Work

Since Geng (2016) formally formulated the LDL problem and introduced a dedicated LDL algorithm, a substantial body of research has emerged to design algorithms tailored to the characteristics of LDL. Among these studies, the most prevalent direction focused on designing methods to capture the intrinsic correlations within the data. For example, Jia et al. (2018) proposed to preserve global label correlations by constraining parameter distances. Zhao & Zhou (2018) proposed to capture correlation by optimal transport. Zheng et al. (2018); Jia et al. (2019b) explored local label correlations, while Ren et al. (2019b) simultaneously focused on global and local label correlations. Ren et al. (2019a) proposed a novel correlation assumption linking label correlations to output similarity. Wang & Geng (2018; 2019) proposed to integrate label distribution information into binary encoding process for efficient nearest neighbor retrieval. Wang & Geng (2023) proposed to capture the label correlation by learning the manifold structure of label distributions. Wang et al. (2025) utilized fuzzy clustering to capture multiple local label correlations. Kou et al. (2024b) introduced an auxiliary multi-label learning process within the LDL framework, and focused on capturing low-rank label correlation within the auxiliary multi-label data. Additionally, another line of research has focused on designing specialized loss functions for non-conventional label distribution scenarios. For example, various machine learning techniques have been introduced for incomplete label distributions, such as low-rank matrix completion (Xu & Zhou, 2017), manifold regularization (Jia et al., 2019a), neighbor-based reconstruction (Zeng et al., 2019; 2020). Various denoising LDL algorithms have been proposed to address the inaccurate label distribution supervision (Kou et al., 2023; Lu et al., 2025a; Kou et al., 2024a; 2025a). Wu et al. (2025) proposed a sub-task loss functions to integrate LDL with more learning tasks related to LDL. González et al. (2021) adapted the one-vs-one decomposition strategy to LDL framework to mitigate issues like class imbalance, noise, and overlapping. Li et al. (2025) introduced the concept of "approximate correctness" to serve as both a more discriminative evaluation metric and a robust learning objective for LDL, addressing the insensitivity of existing metrics and objectives' susceptibility to overfitting. Lu et al. (2025b) proposed an inter-anchor angular regularization term to address the entropy bias problem in LDL task. LDL algorithms based on binary labels (Lu et al., 2023b; Xu et al., 2021; Lu et al., 2025c), ternary labels (Lu & Jia, 2024), or label rankings (Lu & Jia, 2022; Lu et al., 2023a) have been proposed to address the availability of label distribution.

Recent research has increasingly focused on the practical applicability of LDL methods in real-world decision-making. For instance, Wang & Geng (2021); Wang et al. (2022) highlighted and addressed a fundamental mismatch when

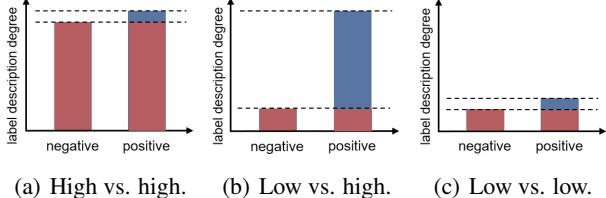

Figure 3. Representative cases of varying degrees of divisiveness.

applying LDL to classification tasks: LDL aims to minimize global distributional difference, whereas classification primarily concerns the label with the highest description degree. Following this direction, Jia et al. (2023b;a); Kou et al. (2025b) extended the concept from the highest description degree to label ranking, and identified the misalignment between standard LDL objectives and ranking error, and incorporated specialized ranking error penalties to address this problem. This paper focuses on a critical risk indicator in decision-making: divisiveness, and aims to coordinate the divisiveness error and the global distributional difference.

## 3. Methodology

In this section, we first introduce the formulation of LDL problem. Then, we delve into the formulation and justification of our proposed measurement of divisiveness. Finally, we will elaborate on our proposed method for maintaining divisiveness accuracy during the learning process.

### 3.1. Problem Formulation

Let $\mathcal{X}^d$ denote the $d$-dimensional feature space, $\Delta^m = \{\boldsymbol{v} \in \mathbb{R}_+^m : \|\boldsymbol{v}\|_1 = 1\}$ denote the $m$-dimensional label distribution space. We cope with the training datasets that appear as data pairs $\{(\boldsymbol{x}^{(i)}, \boldsymbol{y}^{(i)})\}_{i=1}^n$, where $\boldsymbol{x}^{(i)} \in \mathcal{X}^d$ and $\boldsymbol{y}^{(i)} = [y_1^{(i)}; y_2^{(i)}; \ldots; y_m^{(i)}] \in \Delta^m$ denote the feature vector and label distribution of the $i$-th instance, respectively. The objective of LDL is to learn a label distribution regressor $f : \mathcal{X}^d \rightarrow \Delta^m$, i.e., a mapping from feature space to label distribution space by the training dataset $\{(\boldsymbol{x}^{(i)}, \boldsymbol{y}^{(i)})\}_{i=1}^n$.

### 3.2. Measurement of Divisiveness

Divisiveness within a label distribution refers to the propensity of a label distribution to exhibit dissension between semantically opposing labels, which can be captured by the relationship of description degree between the semantically opposing labels. As demonstrated in Figure 3, the relationship of description degree can be generally categorized as three cases. 1) Figure 3(a) depicts a scenario where the description degrees of both opposing labels are high. We believe that the divisiveness in this scenario is highest, as the positive and negative labels are evenly matched at a high

description degree. 2) Figure 3(b) exhibits a scenario where the description degree of negative label is low, while the description degree of positive label is high (or vice versa). We believe that the divisiveness in this scenario is relatively low, as the overwhelming dominance of one label significantly limits the capacity of the opposing label to exert substantial dissension. 3) Figure 3(c) presents a scenario where the description degrees of both opposing labels are low. In this case, divisiveness remains low because neither label is sufficiently salient to the instance, despite their balanced description degrees. The lack of prominence in either direction inhibits the emergence of substantial dissension. According to the aforementioned case study, it can be derived that the divisiveness between the opposing labels is positively related to the minimum of the description degree values of the two labels, i.e., the red portion in Figure 3. Hence, we can utilize $\min\{u, v\}$ to measure the divisiveness between two opposing labels with description degrees $u$ and $v$.

From a crowdsourcing perspective, let $K$ denote the total number of voters, then the description degree $u$ of a label indicates that $K \cdot u$ individuals support voting for that label. Assuming that each supporter of the positive label can form a potential conflict relationship with only one supporter of the negative label, the divisiveness $\min\{u, v\}$ can be practically interpreted as follows: among the $K$ voters, $K \cdot \min\{u, v\}$ positive label supporters are in potential conflict with the same number of negative label supporters. Formally, $\min\{u, v\}$ can also be rewritten as $\min\{u, v\} = \frac{1}{2}(u + v - |u - v|)$. It is evident that if the description degree difference between two opposing labels (i.e., $|u - v|$) is fixed, the larger description degree of any label will lead to the higher divisiveness; if the sum of the description degrees of two opposing labels (i.e., $u + v$) is fixed, the smaller difference in description degree will lead to the higher divisiveness, which is consistent with the intuition in Figures 1 and 3.

More generally, we categorize all labels into two semantically opposing groups, namely the positive and negative label groups[1], thereby enabling the definition of the overall divisiveness within a label distribution. We first introduce the concept of polarity vectors, and then formally define the overall divisiveness within a label distribution.

**Definition 3.1** (Polarity Vectors). The positivity vector $\boldsymbol{\rho}$ and negativity vector $\boldsymbol{\eta}$ are $m$-dimensional vectors defined as $\boldsymbol{\rho} = [\rho_1; \rho_2; \ldots; \rho_m]$ and $\boldsymbol{\eta} = [\eta_1; \eta_2; \ldots; \eta_m]$. Each component $\rho_i$ of $\boldsymbol{\rho}$ represents the degree of positivity associated with the $i$-th label, satisfying $0 \leq \rho_i \leq 1$ for any $i \in \{1, 2, \ldots, m\}$. $\rho_i = 1$ indicates that the label is fully positive in semantics, whereas $\rho_i = 0$ indicates the absence of positivity. Each component $\eta_i$ of $\boldsymbol{\eta}$ represents the de-

---

gree of negativity associated with the $i$-th label, satisfying $0 \leq \eta_i \leq 1$ for all $i \in \{1, 2, \ldots, m\}$. $\eta_i = 1$ indicates that the label is fully negative in semantics, whereas $\eta_i = 0$ indicates the absence of negativity. Besides, we assume $\{i \mid 1 \leq i \leq m, (\rho_i > 0) \wedge (\eta_i > 0)\} = \varnothing$, indicating that no label can be both positive and negative simultaneously.

[1]Without loss of generality, the adjectives "positive" and "negative" are employed to distinctively describe a pair of semantically opposing labels (or label groups) throughout this paper.

**Definition 3.2** (Divisiveness within Label Distribution). Given a positivity vector $\boldsymbol{\rho}$ and a negativity vector $\boldsymbol{\eta}$, the divisiveness within the label distribution $\boldsymbol{y}$ is defined by:

$$\psi(\boldsymbol{y}; \boldsymbol{\rho}, \boldsymbol{\eta}) = \min\{\langle \boldsymbol{\rho}, \boldsymbol{y} \rangle, \langle \boldsymbol{\eta}, \boldsymbol{y} \rangle\}, \quad (1)$$

where $\langle \cdot, \cdot \rangle$ denotes the inner product of two vectors.

**Theoretical Justification.** Although Definition 3.2 has a certain intuitive appeal, it must be acknowledged that multiple plausible definitions could also be conceived based on intuition alone. For example, the divisiveness measure can also be intuitively defined as $\varphi(\boldsymbol{y}; \boldsymbol{\rho}, \boldsymbol{\eta}) = \sum_{i=1}^{m} \sum_{j=1}^{m} \rho_i \cdot \eta_j \cdot \min\{y_i, y_j\}$. To theoretically validate the feasibility or superiority of each plausible definition, a crucial step is to clarify the fundamental properties that a divisiveness measure should possess. To this end, we propose polarity monotonicity as an axiomatic property of divisiveness.

**Definition 3.3** (Redistribution). Let $h(\boldsymbol{y}; \boldsymbol{\rho}, \boldsymbol{\eta})$ denote a function for measuring the divisiveness within a label distribution $\boldsymbol{y}$ under positivity vector $\boldsymbol{\rho}$ and negativity vector $\boldsymbol{\eta}$. Considering an operation that transfers a mass $\delta > 0$ from the $i$-th label to the $j$-th label and yields a new label distribution $\boldsymbol{y}^{\{i \to j\}} = [y_1; \ldots; y_i - \delta; \ldots; y_j + \delta; \ldots; y_m] \in \Delta^m$, the vector $\boldsymbol{y}^{\{i \to j\}}$ is referred to as the redistribution of $\boldsymbol{y}$. The redistribution is *positively directed* if $\langle \boldsymbol{y}^{\{i \to j\}}, \boldsymbol{\rho} \rangle > \langle \boldsymbol{y}^{\{i \to j\}}, \boldsymbol{\eta} \rangle$; otherwise, it is called to be *negatively directed*.

**Definition 3.4** (Polarity Monotonicity). Let $h(\boldsymbol{y}; \boldsymbol{\rho}, \boldsymbol{\eta})$ denote a function for measuring the divisiveness within a label distribution $\boldsymbol{y}$ under positivity vector $\boldsymbol{\rho}$ and negativity vector $\boldsymbol{\eta}$. $h(\boldsymbol{y}; \boldsymbol{\rho}, \boldsymbol{\eta})$ is called to preserve the *polarity monotonicity*, if $h(\boldsymbol{y}; \boldsymbol{\rho}, \boldsymbol{\eta}) \leq h(\boldsymbol{y}^{\{i \to j\}}; \boldsymbol{\rho}, \boldsymbol{\eta})$ holds for any negatively directed redistribution $\boldsymbol{y}^{\{i \to j\}}$ and $\rho_i \leq \rho_j$, or for any positively directed redistribution $\boldsymbol{y}^{\{i \to j\}}$ and $\eta_i \leq \eta_j$.

Definition 3.4 formally captures a fundamental axiom that the divisiveness within a label distribution will not decrease if the weight (aggregate description degree) of either the positive or negative label group increases. Based on Definition 3.4, we theoretically analyze the rationale behind different definitions of divisiveness measure.

**Theorem 3.1.** *Given a label distribution $\boldsymbol{y}$, a positivity vector $\boldsymbol{\rho}$, and a negativity vector $\boldsymbol{\eta}$, the divisiveness measure $\psi(\boldsymbol{y}; \boldsymbol{\rho}, \boldsymbol{\eta})$ defined by Definition 3.2 preserves the polarity monotonicity.*

Theorem 3.1, whose proof is provided in Appendix A.1, theoretically justifies the superiority of Definition 3.2 in

measuring divisiveness. In terms of the plausible alternative measure $\varphi$ and other conventional LDL evaluation metrics, we prove that they cannot preserve the monotonicity:

**Proposition 3.1.** *Given a label distribution $\boldsymbol{y}$, a positivity vector $\boldsymbol{\rho}$, and a negativity vector $\boldsymbol{\eta}$, the divisiveness measure $\varphi(\boldsymbol{y}; \boldsymbol{\rho}, \boldsymbol{\eta}) = \sum_{i=1}^{m} \sum_{j=1}^{m} \rho_i \cdot \eta_j \cdot \min\{y_i, y_j\}$ cannot preserve the polarity monotonicity. Besides, the conventional LDL evaluation metrics Chebyshev distance, Clark distance, Canberra metric, KL divergence, cosine coefficient, intersection similarity (Geng, 2016), Spearman's rank coefficient of correlation (Jia et al., 2023b), $\mu$ metric (Li et al., 2025) cannot preserve the polarity monotonicity*

The proof of Proposition 3.1 can be found in Appendix A.2. Finally, the divisiveness error of the predicted label distribution $\tilde{\boldsymbol{y}}$ against the ground-truth $\boldsymbol{y}$ can be defined by:

$$\mathrm{DVSE}(\tilde{\boldsymbol{y}}, \boldsymbol{y}; \boldsymbol{\rho}, \boldsymbol{\eta}) = |\psi(\tilde{\boldsymbol{y}}; \boldsymbol{\rho}, \boldsymbol{\eta}) - \psi(\boldsymbol{y}; \boldsymbol{\rho}, \boldsymbol{\eta})|. \quad (2)$$

### 3.3. Learning with Divisiveness

Intuitively, conventional LDL loss functions can inherently preserve divisiveness information to some extent, since minimizing the empirical loss toward zero also drives the corresponding divisiveness error toward zero. However, the omnipresence of data noise typically prevents the training loss from reaching zero. When the empirical error based on conventional loss functions remains non-zero, the underlying divisiveness error can span a wide range, encompassing both favorable and unfavorable divisiveness predictions. Figure 2 has visually described this phenomenon, and the following proposition provides a theoretical explanation.

**Theorem 3.2** (Divisiveness Error Bound). *Let $\boldsymbol{y}, \tilde{\boldsymbol{y}} \in \Delta^m$ denote the $m$-dimensional ground-truth and predicted label distributions, respectively. Let $\psi(\boldsymbol{y}; \boldsymbol{\rho}, \boldsymbol{\eta})$ denote the divisiveness-measuring function defined by Equation (1), where $\boldsymbol{\rho}$ and $\boldsymbol{\eta}$ denote the positive and negative vectors, respectively. The divisiveness error of the predicted label distribution against the ground-truth is bounded by*

$$
\begin{aligned}
0 \leq |\psi(\boldsymbol{y}; \boldsymbol{\rho}, \boldsymbol{\eta}) - \psi(\tilde{\boldsymbol{y}}; \boldsymbol{\rho}, \boldsymbol{\eta})| &\leq \xi_\infty \cdot \sqrt{2\mathcal{D}_{\mathrm{KL}}(\tilde{\boldsymbol{y}} \| \boldsymbol{y})}, \\
0 \leq |\psi(\boldsymbol{y}; \boldsymbol{\rho}, \boldsymbol{\eta}) - \psi(\tilde{\boldsymbol{y}}; \boldsymbol{\rho}, \boldsymbol{\eta})| &\leq \xi_\infty \cdot \|\boldsymbol{y} - \tilde{\boldsymbol{y}}\|_1, \\
0 \leq |\psi(\boldsymbol{y}; \boldsymbol{\rho}, \boldsymbol{\eta}) - \psi(\tilde{\boldsymbol{y}}; \boldsymbol{\rho}, \boldsymbol{\eta})| &\leq \xi_2 \cdot \|\boldsymbol{y} - \tilde{\boldsymbol{y}}\|_2,
\end{aligned}
$$
$$(3)$$

*where $\mathcal{D}_{\mathrm{KL}}(\tilde{\boldsymbol{y}} \| \boldsymbol{y})$ is the KL divergence between $\boldsymbol{y}$ and $\tilde{\boldsymbol{y}}$; $\|\cdot\|_1$ and $\|\cdot\|_2$ are the $L_1$ and $L_2$ vector norm, respectively; $\xi_\infty = \max\{\|\boldsymbol{\rho}\|_\infty, \|\boldsymbol{\eta}\|_\infty\}$, and $\xi_2 = \max\{\|\boldsymbol{\rho}\|_2, \|\boldsymbol{\eta}\|_2\}$.*

The proof is provided in Appendix A.4. Theorem 3.2 reveals that conventional LDL loss cannot fully account for divisiveness. In particular, when the MSE or KL divergence is small (approaching a converged state), the square root operation can cause the range of divisiveness errors to expand. In order to explicitly preserve divisiveness information, we

aim to design a dedicated divisiveness loss term, denoted as $\mathcal{L}_\psi$ in this subsection. The final objective function is then formulated as a composite of $\mathcal{L}_\psi$ and the traditional KL or MSE loss. The most straightforward formulation for $\mathcal{L}_\psi$ can be the divisiveness error given in Equation (2) or its squared variant, since the divisiveness measure in Definition 3.2 is Lipschitz continuous w.r.t. the label distribution:

**Proposition 3.2** (Lipschitz Continuity of Divisiveness Measure). *Given a label distribution $\boldsymbol{y}$, a positivity vector $\boldsymbol{\rho}$, and a negativity vector $\boldsymbol{\eta}$, the divisiveness measure $\psi(\boldsymbol{y}; \boldsymbol{\rho}, \boldsymbol{\eta})$ defined by Definition 3.2 is $C$-Lipschitz continuous w.r.t. the label distribution $\boldsymbol{y}$ for any $\|\cdot\|_q$ norm with $p^{-1} + q^{-1} = 1$, where Lipschitz constant $C = \max\{\|\boldsymbol{\rho}\|_p, \|\boldsymbol{\eta}\|_p\}$. Moreover, the Lipschitz constant $C$ is tight, in the sense that it cannot be improved in general.*

The proof is provided in Appendix A.3. Accordingly, the divisiveness loss can be preliminarily expressed as follows:

$$\mathcal{L}_\psi = \mathcal{E}\big(\psi(\tilde{\boldsymbol{y}}; \boldsymbol{\rho}, \boldsymbol{\eta}) - \psi(\boldsymbol{y}; \boldsymbol{\rho}, \boldsymbol{\eta})\big), \quad (4)$$

where $\mathcal{E}(\cdot)$ is an error function such as absolute error $|\cdot|$ or squared error $(\cdot)^2$. However, Equation (4) yields identical gradient directions for all semantically parallel labels. Taking the case where $\mathcal{E}(\cdot) = |\cdot|$ as an example, the gradient of Equation (4) w.r.t. the predicted label distribution satisfies

$$\nabla_{\tilde{\boldsymbol{y}}} |\psi(\tilde{\boldsymbol{y}}; \boldsymbol{\rho}, \boldsymbol{\eta}) - \psi(\boldsymbol{y}; \boldsymbol{\rho}, \boldsymbol{\eta})| \in \{\pm\boldsymbol{\rho}, \pm\boldsymbol{\eta}\}, \quad (5)$$

where $\boldsymbol{\rho}$ and $\boldsymbol{\eta}$ are non-negative vectors. However, the gradient directions for individual labels inherently differ, since increasing the description degree of one label necessarily entails decreasing the description degrees of one or more other labels to maintain the simplex constraint of label distributions. Although the KL or MSE can partially compensate for such gradient inconsistency, they may still hinder the convergence efficiency to some extent, or even lead to model non-convergence. Therefore, we propose a pairwise divisiveness loss function as a surrogate for divisiveness error:

$$\hat{\mathcal{L}}_\psi = \frac{1}{m(m-1)} \sum_{1 \leq i \neq j \leq m} \mathcal{E}(\psi(\tilde{\boldsymbol{y}}; \boldsymbol{\rho} \odot \boldsymbol{e}_i, \boldsymbol{\eta} \odot \boldsymbol{e}_j) - \psi_{ij}^\star), \quad (6)$$

where $\mathcal{E}(\cdot)$ is a error function such as squared error and absolute error, $\boldsymbol{e}_i$ denotes the one-hot vector with the $i$-th entry being 1, $\psi_{ij}^\star = \psi(\boldsymbol{y}; \boldsymbol{\rho} \odot \boldsymbol{e}_i, \boldsymbol{\eta} \odot \boldsymbol{e}_j)$ denotes the pairwise divisiveness computed from the ground-truth label distribution. Unlike Equation (4), the pairwise decomposition mechanism determines individualized gradient directions for each label according to the divisiveness of each distinct label pair. In terms of the consistency between $\mathcal{L}_\psi$ and $\hat{\mathcal{L}}_\psi$, the surrogate loss $\hat{\mathcal{L}}_\psi$ corresponds to a fully enumerated, second-order U-statistic of $\mathcal{L}_\psi$ defined over all label pairs, which implies that the proposed surrogate loss constitutes an unbiased estimator of the original divisiveness error, and

minimizing the surrogate divisiveness loss simultaneously encourages the minimization of original divisiveness error.

Since the surrogate loss function $\hat{\mathcal{L}}_\psi$ is not globally differentiable due to the $\min(\cdot, \cdot)$ operation, we employ the following globally differentiable function $\phi$ to approximate the divisiveness measure $\psi$. Function $\phi$ is formalized as:

$$\phi(\tilde{\boldsymbol{y}}; \boldsymbol{\rho}, \boldsymbol{\eta}) = \big( \exp\left(-k\langle\boldsymbol{\rho}, \tilde{\boldsymbol{y}}\rangle\right) + \exp\left(-k\langle\boldsymbol{\eta}, \tilde{\boldsymbol{y}}\rangle\right) \big)^{-1} \times \big(\langle\boldsymbol{\rho}, \tilde{\boldsymbol{y}}\rangle \exp\left(-k\langle\boldsymbol{\rho}, \tilde{\boldsymbol{y}}\rangle\right) + \langle\boldsymbol{\eta}, \tilde{\boldsymbol{y}}\rangle \exp\left(-k\langle\boldsymbol{\eta}, \tilde{\boldsymbol{y}}\rangle\right)\big), \quad (7)$$

where $k > 0$ is a hyper-parameter controlling the function smoothness. In terms of the error function $\mathcal{E}$, we employ a smoothed variant of the absolute error $\mathcal{E}(\cdot) = \sqrt{(\cdot)^2 + \varepsilon^2}$. Finally, the overall loss function can be summarized as:

$$\mathcal{L} = \frac{1}{n} \sum_{i=1}^{n} \mathcal{D}_{\mathrm{KL}}^{(i)} + \alpha \hat{\mathcal{L}}_\phi^{(i)}, \quad (8)$$

where $\alpha$ is a trade-off hyper-parameter; $\mathcal{D}_{\mathrm{KL}}^{(i)}$ and $\hat{\mathcal{L}}_\phi^{(i)}$ are the KL divergence and the surrogate divisiveness loss with global differentiability for the $i$-th instance, respectively.

## 4. Experiments

### 4.1. Experimental Setup

**Datasets and Divisiveness Settings.** We carry out experiments on the following widely-used benchmarks: RAF-ML (Li & Deng, 2019), Music (Lee et al., 2021), Flickr, Twitter, and emotion6 provided by (Yang et al., 2017), M²B (Nguyen et al., 2012) and Painting (Machajdik & Hanbury, 2010). For datasets with ordinal label distributions, we construct divisiveness settings using a staircase scheme. For example, in the 5-label M²B dataset, we set $\boldsymbol{\rho} = [0.0, 0.0, 0.0, 0.5, 1.0]$ and $\boldsymbol{\rho} = [1.0, 0.5, 0.0, 0.0, 0.0]$. For datasets with discrete categories, we instead partition labels into positive and negative sets according to their semantic polarity. If the $j$-th label belongs to the positive set, we set $\rho_j = 1$ and $\eta_j = 0$; if it belongs to the negative set, we set $\rho_j = 0$ and $\eta_j = 1$; otherwise, we set $\rho_j = \eta_j = 0$. Concretely, for facial emotion recognition datasets RAF-ML and emotion6, *happiness* and *surprise* are treated as positive labels, whereas *sadness*, *anger*, *disgust*, and *fear* are treated as negative labels; *neutral* is excluded from both sets. For image emotion distribution datasets Flickr, Twitter, and Painting, *amusement*, *contentment*, and *excitement* are positive labels, while *anger*, *disgust*, *fear*, and *sadness* are negative labels; *awe* is excluded. For the Music dataset, *calm*, *cheerful*, *danceable*, *love*, and *dreamy* are positive labels, while *tense* and *sad* are negative labels.

**Evaluation Metrics.** We are interested in evaluating how well different methods can preserve the divisiveness of label distributions. Thus, we employ the divisiveness error

(DVSE $\downarrow$) in Equation (2) as a new evaluation metric. We also report several commonly-used metrics in LDL, suggested by Li et al. (2025), including Chebyshev distance (Cheby. $\downarrow$), Clark distance (Clark $\downarrow$), Kullback-Leibler divergence (KLD $\downarrow$), Cosine similarity (Cosine $\uparrow$), Spearman's rank correlation coefficient (Spear. $\uparrow$), and the $\mu \uparrow$ metric. Here $\downarrow$ ($\uparrow$) indicates "the lower (higher) the better".

**Baselines.** We name the method we propose LDL-DVS. We compare LDL-DVS with these methods: SA-BFGS (Geng, 2016), AA-$k$NN (Geng, 2016), LDLLC (Jia et al., 2018), LDLSF (Ren et al., 2019a), LCLR (Ren et al., 2019b), DF-LDL (González et al., 2021), LRR (Jia et al., 2023b), DPA (Jia et al., 2023a), RG4LDL (Tan et al., 2025), and $\delta$-LDL (Li et al., 2025). Except for AA-$k$NN and DF-LDL, all methods adopt a simple feedforward architecture as their backbone model. For LDL-DVS, we fix the hyperparameter $k$ to 10 and $\alpha$ to 0.1 for all datasets. To ensure a fair comparison, we conduct ten-fold experiments repeated 10 times for each method on each dataset.

### 4.2. Results and Discussion

**Comparison Results.** Representative results are summarized in Table 1, while the remaining results are reported in Appendix A.5. $\bullet$ ($\circ$) indicates "LDL-DVS is statistically superior (inferior) to the comparing methods" (under the pairwise two-tailed $t$-test at 0.05 significance level); there is no significant difference if neither $\bullet$ nor $\circ$ is shown; the best and second-best results are highlighted in **bold** and underline, respectively. It can be observed that, across all datasets except Painting, LDL-DVS consistently achieves the best performance in terms of both DVSE and KLD, with particularly notable improvements on the RAF-ML and Music datasets. These results indicate that DVSE and KLD can be complementary to each other, and empirically validate our motivation of introducing $\hat{\mathcal{L}}_\phi(\cdot, \cdot)$ as an auxiliary term to improve the learning process of label distributions.

Two additional observations are worth noting. First, DVSE serves as a unique evaluation metric: by deliberately ranking comparison methods according to DVSE, we find that it captures aspects of label predictions that are not observable with conventional metrics (i.e., divisiveness consistency). Second, the performance advantage of LDL-DVS in terms of DVSE may come at the expense of modeling the sparsity of the label distributions. This trade-off can be observed in its relatively lower performance on the Clark metric.

We also observe certain limitations. For instance, on the Painting dataset, some baseline methods, although not optimizing $\hat{\mathcal{L}}_\phi(\cdot, \cdot)$, may obtain smaller DVSE values than LDL-DVS. This phenomenon can be attributed to two aspects: 1) These methods tend to produce more uniform label distribution predictions, which coincidentally leads to

*Table 1.* Experimental results on `RAF-ML`, `Music`, `Painting`, and `emotion6` formatted as "mean $\pm$ std".

| Algorithms | Cheby. ↓ | Clark ↓ | KLD ↓ | Cosine ↑ | Spear. ↑ | $\mu$ ↑ | DVSE ↓ |
|---|---|---|---|---|---|---|---|
| | | | **RAF-ML** | | | | |
| AA-$k$NN | ● .2643$_{\pm.005}$ | ● 1.4882$_{\pm.014}$ | ● .4073$_{\pm.014}$ | ● .8221$_{\pm.006}$ | ● .6041$_{\pm.013}$ | ● 49.37%$_{\pm.012}$ | ● .1170$_{\pm.004}$ |
| RG4LDL | ● .3697$_{\pm.005}$ | ● 1.5868$_{\pm.012}$ | ● .6784$_{\pm.015}$ | ● .6788$_{\pm.007}$ | ● .2841$_{\pm.024}$ | ● 19.13%$_{\pm.010}$ | ● .1118$_{\pm.004}$ |
| LDLSF | ● .1885$_{\pm.004}$ | ● 1.5429$_{\pm.013}$ | ● .8256$_{\pm.069}$ | ● .9102$_{\pm.005}$ | ● .7713$_{\pm.010}$ | ● 48.58%$_{\pm.017}$ | ● .1036$_{\pm.003}$ |
| DF-LDL | ● .1546$_{\pm.005}$ | ○ 1.4196$_{\pm.015}$ | ● .1957$_{\pm.011}$ | ● .9273$_{\pm.005}$ | ● .7817$_{\pm.010}$ | ● 75.68%$_{\pm.010}$ | ● .0997$_{\pm.004}$ |
| LCLR | ● .1538$_{\pm.005}$ | ○ 1.4199$_{\pm.015}$ | ● .1957$_{\pm.011}$ | ● .9277$_{\pm.005}$ | ● .7839$_{\pm.010}$ | ● 75.77%$_{\pm.010}$ | ● .0990$_{\pm.003}$ |
| SA-BFGS | ● .1538$_{\pm.005}$ | ○ 1.4199$_{\pm.015}$ | ● .1957$_{\pm.011}$ | ● .9277$_{\pm.005}$ | ● .7840$_{\pm.010}$ | ● 75.77%$_{\pm.010}$ | ● .0986$_{\pm.003}$ |
| LDLLC | ● .1538$_{\pm.005}$ | ○ 1.4178$_{\pm.015}$ | ● .1956$_{\pm.011}$ | ● .9276$_{\pm.005}$ | ● .7842$_{\pm.010}$ | ● 75.79%$_{\pm.010}$ | ● .0985$_{\pm.003}$ |
| DPA | ● .1538$_{\pm.005}$ | ○ 1.4185$_{\pm.016}$ | ● .1957$_{\pm.011}$ | ● .9276$_{\pm.005}$ | ● .7841$_{\pm.010}$ | ● 75.80%$_{\pm.010}$ | ● .0985$_{\pm.003}$ |
| LRR | ● .1538$_{\pm.005}$ | ○ 1.4185$_{\pm.015}$ | ● .1957$_{\pm.011}$ | ● .9276$_{\pm.005}$ | ● .7841$_{\pm.010}$ | ● 75.79%$_{\pm.010}$ | ● .0984$_{\pm.003}$ |
| $\delta$-LDL | ● .1527$_{\pm.005}$ | ○ **1.4159**$_{\pm.016}$ | ● .1929$_{\pm.011}$ | ● .9284$_{\pm.005}$ | ● .7847$_{\pm.010}$ | ● 76.06%$_{\pm.010}$ | ● .0829$_{\pm.004}$ |
| LDL-DVS | **.1493**$_{\pm.005}$ | 1.4210$_{\pm.016}$ | **.1873**$_{\pm.011}$ | **.9300**$_{\pm.005}$ | **.7863**$_{\pm.009}$ | **76.67%**$_{\pm.010}$ | **.0764**$_{\pm.003}$ |
| | | | **Music** | | | | |
| DF-LDL | ● .1022$_{\pm.008}$ | ● .8489$_{\pm.057}$ | ● .1719$_{\pm.026}$ | ● .8726$_{\pm.018}$ | ● .2688$_{\pm.080}$ | ● 26.18%$_{\pm.045}$ | ● .0866$_{\pm.013}$ |
| SA-BFGS | ● .1020$_{\pm.008}$ | ● .8802$_{\pm.060}$ | ● .1912$_{\pm.030}$ | ● .8665$_{\pm.018}$ | ● .3030$_{\pm.072}$ | ● 25.57%$_{\pm.047}$ | ● .0808$_{\pm.010}$ |
| LCLR | ● .1000$_{\pm.008}$ | ● .8644$_{\pm.064}$ | ● .1821$_{\pm.034}$ | ● .8738$_{\pm.019}$ | ● .3673$_{\pm.073}$ | ● 26.23%$_{\pm.049}$ | ● .0804$_{\pm.012}$ |
| DPA | ● .0971$_{\pm.007}$ | ● .8532$_{\pm.059}$ | ● .1758$_{\pm.027}$ | ● .8746$_{\pm.017}$ | ● .3224$_{\pm.072}$ | ● 27.04%$_{\pm.047}$ | ● .0791$_{\pm.011}$ |
| LRR | ● .0969$_{\pm.007}$ | ● .8519$_{\pm.058}$ | ● .1751$_{\pm.027}$ | ● .8750$_{\pm.017}$ | ● .3214$_{\pm.071}$ | ● 27.11%$_{\pm.047}$ | ● .0785$_{\pm.010}$ |
| LDLLC | ● .0968$_{\pm.007}$ | ● .8518$_{\pm.057}$ | ● .1751$_{\pm.027}$ | ● .8751$_{\pm.017}$ | ● .3212$_{\pm.070}$ | ● 27.12%$_{\pm.046}$ | ● .0776$_{\pm.011}$ |
| LDLSF | ● .0946$_{\pm.006}$ | ● .9498$_{\pm.072}$ | ● .6168$_{\pm.306}$ | ● .8718$_{\pm.018}$ | ● .3123$_{\pm.072}$ | ● 23.89%$_{\pm.045}$ | ● .0774$_{\pm.009}$ |
| AA-$k$NN | ● .0784$_{\pm.006}$ | ● .7499$_{\pm.061}$ | ● .1258$_{\pm.021}$ | ● .9067$_{\pm.015}$ | ● .3989$_{\pm.081}$ | ● 36.75%$_{\pm.045}$ | ● .0711$_{\pm.008}$ |
| RG4LDL | ● .0874$_{\pm.005}$ | ● .7827$_{\pm.052}$ | ● .1304$_{\pm.017}$ | ● .9023$_{\pm.011}$ | ● .3420$_{\pm.064}$ | ● 30.49%$_{\pm.037}$ | ● .0674$_{\pm.008}$ |
| $\delta$-LDL | **.0733**$_{\pm.005}$ | ● .7185$_{\pm.055}$ | .1061$_{\pm.016}$ | .9218$_{\pm.011}$ | .4942$_{\pm.058}$ | ● 39.35%$_{\pm.036}$ | .0573$_{\pm.008}$ |
| LDL-DVS | **.0733**$_{\pm.005}$ | **.7135**$_{\pm.053}$ | **.1052**$_{\pm.015}$ | **.9225**$_{\pm.010}$ | **.4965**$_{\pm.052}$ | **39.90%**$_{\pm.033}$ | **.0571**$_{\pm.007}$ |
| | | | **Painting** | | | | |
| $\delta$-LDL | .2484$_{\pm.020}$ | **1.7019**$_{\pm.067}$ | .5288$_{\pm.048}$ | .7377$_{\pm.023}$ | .3146$_{\pm.092}$ | 23.05%$_{\pm.037}$ | .1574$_{\pm.028}$ |
| DPA | ● .2585$_{\pm.020}$ | ● 1.7561$_{\pm.069}$ | ● .5999$_{\pm.067}$ | ● .7116$_{\pm.028}$ | ● .2780$_{\pm.069}$ | ● 20.48%$_{\pm.041}$ | .1529$_{\pm.019}$ |
| LDLLC | ● .2586$_{\pm.020}$ | ● 1.7560$_{\pm.069}$ | ● .5994$_{\pm.066}$ | ● .7120$_{\pm.027}$ | ● .2774$_{\pm.068}$ | ● 20.48%$_{\pm.041}$ | .1527$_{\pm.016}$ |
| LDLSF | ● .2853$_{\pm.023}$ | ● 1.8290$_{\pm.067}$ | ● 4.1107$_{\pm1.205}$ | ● .6565$_{\pm.033}$ | ● .2047$_{\pm.067}$ | ● 11.89%$_{\pm.040}$ | .1526$_{\pm.021}$ |
| LRR | ● .2587$_{\pm.019}$ | ● 1.7556$_{\pm.068}$ | ● .5988$_{\pm.064}$ | ● .7118$_{\pm.027}$ | ● .2782$_{\pm.068}$ | ● 20.48%$_{\pm.040}$ | .1501$_{\pm.017}$ |
| RG4LDL | ● .2974$_{\pm.024}$ | ● 1.8273$_{\pm.067}$ | ● .8089$_{\pm.126}$ | ● .6501$_{\pm.033}$ | ● .1870$_{\pm.070}$ | ● 15.65%$_{\pm.038}$ | .1478$_{\pm.018}$ |
| LCLR | ● .2979$_{\pm.028}$ | ● 1.8632$_{\pm.070}$ | ● .9065$_{\pm.215}$ | ● .6579$_{\pm.036}$ | ● .2454$_{\pm.066}$ | ● 16.23%$_{\pm.045}$ | .1471$_{\pm.019}$ |
| SA-BFGS | ● .2965$_{\pm.028}$ | ● 1.8559$_{\pm.073}$ | ● .8997$_{\pm.233}$ | ● .6596$_{\pm.036}$ | ● .2363$_{\pm.069}$ | ● 16.37%$_{\pm.044}$ | ○ .1468$_{\pm.019}$ |
| DF-LDL | ● .2714$_{\pm.021}$ | ● 1.8051$_{\pm.064}$ | ● .6950$_{\pm.096}$ | ● .6772$_{\pm.032}$ | ● .2310$_{\pm.073}$ | ● 17.13%$_{\pm.044}$ | ○ .1445$_{\pm.018}$ |
| AA-$k$NN | ● .2572$_{\pm.021}$ | ● 1.7553$_{\pm.060}$ | ● .7100$_{\pm.161}$ | ● .6987$_{\pm.027}$ | ● .2241$_{\pm.072}$ | ● 19.76%$_{\pm.041}$ | ○ **.1416**$_{\pm.021}$ |
| LDL-DVS | **.2465**$_{\pm.019}$ | 1.7028$_{\pm.068}$ | **.5262**$_{\pm.041}$ | **.7389**$_{\pm.019}$ | **.3188**$_{\pm.076}$ | **23.14%**$_{\pm.032}$ | .1523$_{\pm.021}$ |
| | | | **emotion6** | | | | |
| AA-$k$NN | ● .3364$_{\pm.011}$ | ● 1.7139$_{\pm.025}$ | ● .8328$_{\pm.072}$ | ● .6529$_{\pm.013}$ | ● .2257$_{\pm.031}$ | ● 20.59%$_{\pm.015}$ | ● .1222$_{\pm.007}$ |
| LDLSF | ● .3175$_{\pm.009}$ | ● 1.6767$_{\pm.025}$ | ● .9999$_{\pm.123}$ | ● .6997$_{\pm.012}$ | ● .3125$_{\pm.030}$ | ● 23.48%$_{\pm.016}$ | ● .1156$_{\pm.006}$ |
| SA-BFGS | ● .3179$_{\pm.010}$ | ● 1.6705$_{\pm.026}$ | ● .6204$_{\pm.025}$ | ● .6998$_{\pm.012}$ | ● .3257$_{\pm.028}$ | ● 26.14%$_{\pm.014}$ | ● .1151$_{\pm.006}$ |
| LCLR | ● .3234$_{\pm.015}$ | ● 1.6795$_{\pm.030}$ | ● .6454$_{\pm.058}$ | ● .6870$_{\pm.030}$ | ● .2959$_{\pm.070}$ | ● 24.55%$_{\pm.034}$ | ● .1151$_{\pm.008}$ |
| DF-LDL | ● .3168$_{\pm.010}$ | ● 1.6704$_{\pm.026}$ | ● .6162$_{\pm.025}$ | ● .7013$_{\pm.011}$ | ● .3308$_{\pm.028}$ | ● 26.56%$_{\pm.015}$ | ● .1146$_{\pm.006}$ |
| LDLLC | ● .3179$_{\pm.010}$ | 1.6623$_{\pm.026}$ | ● .6129$_{\pm.023}$ | ● .7039$_{\pm.011}$ | ● .3325$_{\pm.030}$ | ● 26.27%$_{\pm.014}$ | ● .1139$_{\pm.006}$ |
| DPA | ● .3215$_{\pm.009}$ | 1.6654$_{\pm.026}$ | ● .6241$_{\pm.023}$ | ● .6970$_{\pm.011}$ | ● .3138$_{\pm.030}$ | ● 25.26%$_{\pm.013}$ | ● .1136$_{\pm.006}$ |
| LRR | ● .3174$_{\pm.010}$ | 1.6621$_{\pm.027}$ | ● .6111$_{\pm.025}$ | ● .7051$_{\pm.011}$ | ● .3323$_{\pm.031}$ | ● 26.36%$_{\pm.014}$ | ● .1133$_{\pm.006}$ |
| RG4LDL | ● .3257$_{\pm.010}$ | 1.6596$_{\pm.026}$ | ● .6206$_{\pm.022}$ | ● .7007$_{\pm.009}$ | ● .3280$_{\pm.027}$ | ● 24.22%$_{\pm.011}$ | ● .1105$_{\pm.006}$ |
| $\delta$-LDL | ● .3031$_{\pm.010}$ | **1.6562**$_{\pm.026}$ | ● .5636$_{\pm.024}$ | ● .7283$_{\pm.012}$ | ● .4051$_{\pm.027}$ | ● 30.97%$_{\pm.016}$ | ● .1083$_{\pm.005}$ |
| LDL-DVS | **.3030**$_{\pm.010}$ | 1.6579$_{\pm.026}$ | **.5624**$_{\pm.024}$ | **.7288**$_{\pm.011}$ | **.4061**$_{\pm.028}$ | **31.26%**$_{\pm.016}$ | **.1074**$_{\pm.005}$ |

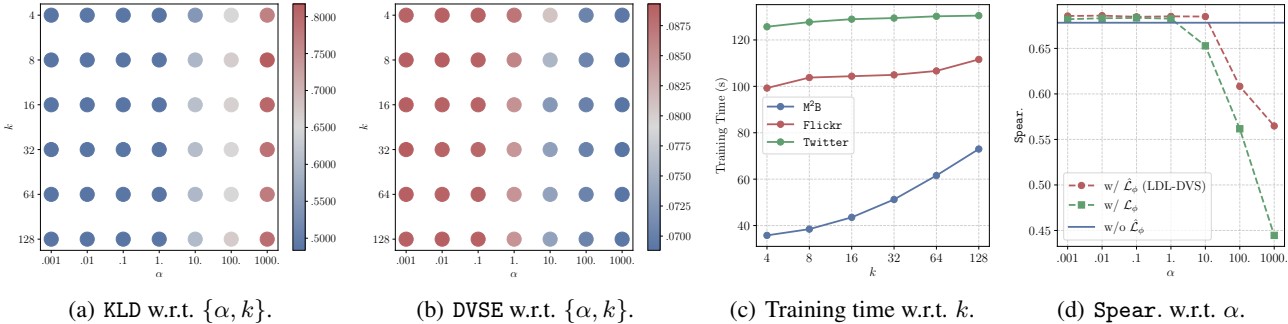

*Figure 4.* Further experimental analysis of LDL-DVS. (a) and (b): parameter sensitivity analysis on the M²B dataset under KLD and DVSE metrics, where blue/red indicates better/worse performance; (c): efficiency study on the M²B dataset; (d): ablation study on the M²B dataset.

smaller DVSE values, but at the cost of prediction accuracy; this trade-off is further corroborated by their inferior performance on other evaluation metrics; 2) None of the compared methods is able to achieve satisfactory performance on the Painting dataset. This may be due to intrinsic issues related to data quality, such as the reliability of extracted features, causing most methods to converge to suboptimal solutions. This observation is further supported by the overall low $\mu$ metric across all comparing algorithms, where the best performance is only 23.14%, achieved by LDL-DVS.

**Parameter Sensitivity.** It can be readily seen that the loss term $\mathcal{L}_\phi(\cdot, \cdot)$ (or $\hat{\mathcal{L}}_\phi(\cdot, \cdot)$) alone induces, in general, a set of minimizers rather than a singleton. Consequently, when the hyperparameter $\alpha$ is set to a large value, the optimization process is dominated by the $\hat{\mathcal{L}}_\phi(\cdot, \cdot)$ term, driving the model to rapidly decrease divisiveness discrepancy. Once a near-minimal DVSE value is reached, the model may continue to move along the corresponding DVSE level set, potentially drifting away from the ground-truth label distribution. Since $\hat{\mathcal{L}}_\phi(\cdot, \cdot)$ does *not* penalize such movements, this behavior may lead to degraded performance on other LDL metrics. To validate this, we conduct a parameter sensitivity analysis over the M²B dataset to identify a reasonable range for $\alpha$. Specifically, we vary $\alpha$ from $10^{-3}$ to $10^3$, and vary $k$ from $2^2$ to $2^7$, and visualize the results in Figures 4(a) and 4(b). The results indicate that $\alpha$ values within $[10^{-2}, 1]$ yield improved KLD performance. We observe that excessively minimizing DVSE may deteriorate other metrics, e.g., KLD, and this finding aligns with our earlier analysis. Additionally, we observe that the parameter $k$ has a negligible impact on performance across metrics. However, interestingly, increasing $k$ consistently incurs a higher computational cost. Particularly, $k = 2^8$ results in NaN values during training, causing the optimization process to fail. The relation between training time and $k$ is illustrated in Figure 4(c), where a monotonic increasing trend can be observed. This is likely attributed to the numerical instability caused by large $k$.

**Ablation Study.** To validate our hypothesis that directly optimizing $\mathcal{L}_\phi(\cdot, \cdot)$ can be misleading for overall label distribution prediction, we conduct an ablation study with the following settings: 1) Replace $\hat{\mathcal{L}}_\phi(\cdot, \cdot)$ with $\mathcal{L}_\phi(\cdot, \cdot)$ (denoted as "w/ $\mathcal{L}_\phi$"); 2) Remove $\hat{\mathcal{L}}_\phi(\cdot, \cdot)$ entirely by setting $\alpha = 0$ (denoted as "w/o $\hat{\mathcal{L}}_\phi$"). The results on the M²B dataset, measured by the Spear. metric, are presented in Figure 4(d). When $\alpha$ is within a moderate range, optimizing $\mathcal{L}_\phi$ achieves performance comparable to optimizing $\hat{\mathcal{L}}_\phi$, both outperforming the baseline model without any additional loss term. As $\alpha$ increases, however, directly optimizing $\mathcal{L}_\phi$ leads to a sharp drop in performance, whereas using the surrogate loss mitigates this degradation. These findings empirically support our previous conjecture regarding the stabilizing effect of the surrogate divisiveness term.

## 5. Conclusion and Limitation Discussion

**Conclusion.** This paper presents a divisiveness-consistent label distribution learning framework to address the critical limitation of conventional methods in preserving the label distribution divisiveness, which is a key indicator of decision risk reflecting the dissension between semantically opposing labels. We first formalize a divisiveness measure satisfying polarity monotonicity and theoretically demonstrate the inherent inconsistency between conventional LDL loss functions and divisiveness error. To overcome the adversarial gradient issue in direct minimization of divisiveness error, we introduce a pairwise divisiveness loss as an unbiased surrogate, enabling stable model convergence. Experiments across multiple real-world datasets demonstrate that our proposed method consistently outperforms baseline approaches on the novel evaluation metric of divisiveness error, while maintaining competitive on other traditional LDL metrics.

**Limitation Discussion.** In terms of the applicability, our proposed divisiveness measure presupposes that the label space contains semantically opposing label pairs. In such scenarios, decision-makers are indeed interested in whether

the label distribution is polarized, making the divisiveness measure a meaningful risk indicator. In contrast, for the tasks without semantic opposition, the polarity vector is an all-zero vector, implying that the degree of divisiveness is also zero. In such scenarios, there is no need for decision-makers to consider divisiveness, as it is invariably absent.

In terms of the loss function, the pairwise divisiveness loss, despite serving as an unbiased estimator for the original divisiveness error, may theoretically provide a weaker approximation when the label space is extremely small. Fortunately, this limitation is of minor practical concern, as most real-world LDL tasks (such as the benchmarks employed in our experiments) typically possess sufficient label cardinality to ensure a satisfactory approximation in practice (e.g. a label cardinality of at least 5). Nevertheless, we intend to address this problem through two research directions in the future. On the one hand, we will theoretically establish the relationship between the fidelity of the surrogate loss and label cardinality. On the other hand, we will explore more effective loss functions that balance the fidelity to original divisiveness error with mitigation of adversarial gradients.

## Impact Statement

This paper presents work whose goal is to advance the field of Machine Learning. There are many potential societal consequences of our work, none which we feel must be specifically highlighted here.

## Acknowledgements

This work was partially supported by the National Natural Science Foundation of China (62476130, 62576166), the Natural Science Foundation of Jiangsu Province (BK20242045), the Innovation and Technology Fund (GHP/079/22SZ, ITS/034/23FP), and the UGC/GRF (No. 15211024, 15215421).

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

# A. Appendix

### A.1. Proof of Theorem 3.1

*Proof.* For convenience, we denote $\hat{\rho} = \langle \boldsymbol{\rho}, \boldsymbol{y} \rangle$ and $\hat{\eta} = \langle \boldsymbol{\eta}, \boldsymbol{y} \rangle$. Equation (1) can be rewritten as $\psi(\boldsymbol{y}; \boldsymbol{\rho}, \boldsymbol{\eta}) = \min\{\hat{\rho}, \hat{\eta}\}$. Under the redistribution $\boldsymbol{y} \to \boldsymbol{y}^{\{i \to j\}}$, the updated summations become $\hat{\rho}^{\{i \to j\}} = \hat{\rho} + \delta(\rho_j - \rho_i)$, $\hat{\eta}^{\{i \to j\}} = \hat{\eta} + \delta(\eta_j - \eta_i)$. We consider the value change $\Delta\psi^{\{i \to j\}} = \psi(\boldsymbol{y}^{\{i \to j\}}; \boldsymbol{\rho}, \boldsymbol{\eta}) - \psi(\boldsymbol{y}; \boldsymbol{\rho}, \boldsymbol{\eta})$, and analyze all possible cases induced by the min operator.

**Case 1:** In this case, we have $\hat{\rho} \leq \hat{\eta}$.

$$\Delta\psi^{\{i \to j\}} = \begin{cases} \hat{\rho}^{\{i \to j\}} - \hat{\rho} = \delta\underbrace{(\rho_j - \rho_i)}_{\rho_i \leq \rho_j} \geq 0 & \text{if } \hat{\rho}^{\{i \to j\}} \leq \hat{\eta}^{\{i \to j\}} \\ \hat{\eta}^{\{i \to j\}} - \hat{\rho} = \underbrace{\hat{\eta} - \hat{\rho}}_{\hat{\rho} \leq \hat{\eta}} + \delta\underbrace{(\eta_j - \eta_i)}_{\eta_i \leq \eta_j} \geq 0 & \text{if } \hat{\rho}^{\{i \to j\}} > \hat{\eta}^{\{i \to j\}} \end{cases} \tag{9}$$

**Case 2:** In this case, we have $\hat{\rho} > \hat{\eta}$.

$$\Delta\psi^{\{i \to j\}} = \begin{cases} \hat{\rho}^{\{i \to j\}} - \hat{\eta} = \underbrace{\hat{\rho} - \hat{\eta}}_{\hat{\rho} > \hat{\eta}} + \delta\underbrace{(\rho_j - \rho_i)}_{\rho_i \leq \rho_j} \geq 0 & \text{if } \hat{\rho}^{\{i \to j\}} \leq \hat{\eta}^{\{i \to j\}} \\ \hat{\eta}^{\{i \to j\}} - \hat{\eta} = \delta\underbrace{(\eta_j - \eta_i)}_{\eta_i \leq \eta_j} \geq 0 & \text{if } \hat{\rho}^{\{i \to j\}} > \hat{\eta}^{\{i \to j\}} \end{cases} \tag{10}$$

Combining all cases, we have $\Delta\psi^{\{i \to j\}} \geq 0$, which completes the proof. $\qquad\square$

### A.2. Proof of Proposition 3.1

*Proof.* Recall that $\varphi(\boldsymbol{y}; \boldsymbol{\rho}, \boldsymbol{\eta})$ is defined as $\varphi(\boldsymbol{y}; \boldsymbol{\rho}, \boldsymbol{\eta}) = \sum_{i=1}^m \sum_{j=1}^m \rho_i \cdot \eta_j \cdot \min\{y_i, y_j\}$. Under the redistribution $\boldsymbol{y} \to \boldsymbol{y}^{\{i \to j\}}$, we denote the value that remains unchanged as $\varphi^{\{i \to j\}} = \sum_{r \notin \{i,j\}} \sum_{r' \notin \{i,j\}} \rho_r \eta_{r'} \min\{y_r, y_{r'}\}$. For convenience, we denote $\hat{\varphi}^{\{i \to j\}} = (\rho_i \eta_j + \rho_j \eta_i) \min\{y_i, y_j\} + \rho_i \eta_i y_i + \rho_j \eta_j y_j$, and $\hat{\varphi}^{\{i \to j\}\prime} = (\rho_i \eta_j + \rho_j \eta_i) \min\{y_i - \delta, y_j + \delta\} + \rho_i \eta_i(y_i - \delta) + \rho_j \eta_j(y_j + \delta)$. Then we have $\varphi(\boldsymbol{y}; \boldsymbol{\rho}, \boldsymbol{\eta}) = \varphi^{\{i \to j\}} + \hat{\varphi}^{\{i \to j\}}$, and $\varphi(\boldsymbol{y}^{\{i \to j\}}; \boldsymbol{\rho}, \boldsymbol{\eta}) = \varphi^{\{i \to j\}} + \hat{\varphi}^{\{i \to j\}\prime}$. We consider the value change

$$\begin{aligned} \Delta\varphi^{\{i \to j\}} &= \varphi(\boldsymbol{y}^{\{i \to j\}}; \boldsymbol{\rho}, \boldsymbol{\eta}) - \varphi(\boldsymbol{y}; \boldsymbol{\rho}, \boldsymbol{\eta}) = \hat{\varphi}^{\{i \to j\}\prime} - \hat{\varphi}^{\{i \to j\}} \\ &= (\rho_i \eta_j + \rho_j \eta_i)(\min\{y_i - \delta, y_j + \delta\} - \min\{y_i, y_j\}) + \delta(\rho_j \eta_j - \rho_i \eta_i), \end{aligned} \tag{11}$$

and analyze all possible cases induced by the min operator.

**Case 1:** In this case, we have $y_i \leq y_j$, which implies $y_i - \delta < y_j + \delta$. Thus we have $\Delta\varphi^{\{i \to j\}} = \delta(\rho_j \eta_j - \rho_i \eta_i - \rho_i \eta_j - \rho_j \eta_i)$.

**Case 2:** In this case, we have $y_i > y_j$. If $y_i - \delta < y_j + \delta$, then

$$\begin{aligned} \Delta\varphi^{\{i \to j\}} &= (y_i - y_j - \delta)(\rho_i \eta_j + \rho_j \eta_i) + \delta(\rho_j \eta_j - \rho_i \eta_i) \\ &= (y_i - y_j)(\rho_i \eta_j + \rho_j \eta_i) + \delta(\rho_j \eta_j - \rho_i \eta_i - \rho_i \eta_j - \rho_j \eta_i). \end{aligned} \tag{12}$$

Otherwise, if $y_i - \delta \geq y_j + \delta$, then $\Delta\varphi^{\{i \to j\}} = \delta(\rho_j \eta_j - \rho_i \eta_i + \rho_i \eta_j + \rho_j \eta_i)$. Combining all cases, we can find that

$$\Delta\varphi^{\{i \to j\}} \gtrless 0 \Leftrightarrow \begin{cases} b - a \gtrless 0, & \Delta y > 0, \\ \delta(b - a) - a\Delta y \gtrless 0, & -2\delta < \Delta y \leq 0, \\ a + b \gtrless 0, & \Delta y \leq -2\delta, \end{cases} \tag{13}$$

where $a = \rho_i \eta_j + \rho_j \eta_i \geq 0$, $b = \rho_j \eta_j - \rho_i \eta_i$, and $\Delta y = y_j - y_i$. Since these conditions are unguaranteed to hold in general, by choosing different $\boldsymbol{\rho}$, $\boldsymbol{\eta}$, and $\delta$, we can always find counterexamples such that $\Delta\varphi^{\{i \to j\}}$ is either positive or negative.

The remaining metrics to be analyzed do not involve $\boldsymbol{\rho}$ and $\boldsymbol{\eta}$, and thus are incapable of capturing the divisiveness, i.e., under the redistribution $\boldsymbol{y} \to \boldsymbol{y}^{\{i \to j\}}$, these metrics may either increase or decrease. For example, the monotonicity of KL divergence requires the following conditions to hold:

$$\Delta \mathcal{D}_{\mathrm{KL}}{}^{\{i \to j\}} \gtrless 0 \Leftrightarrow y_j \log \frac{\tilde{y}_j}{\tilde{y}_j + \delta} + y_i \log \frac{\tilde{y}_i}{\tilde{y}_i - \delta} \gtrless 0, \tag{14}$$

which are not guaranteed to hold in general as well. Proofs for the remaining metrics are omitted for brevity. □

### A.3. Proof of Proposition 3.2

*Proof.* For convenience, we denote $\hat{\rho}_r = \langle \boldsymbol{\rho}, \boldsymbol{y}_r \rangle$ and $\hat{\eta}_r = \langle \boldsymbol{\eta}, \boldsymbol{y}_r \rangle$ for any label distribution $\boldsymbol{y}_r$. Let $\boldsymbol{y}^{(i)}$ and $\boldsymbol{y}^{(j)}$ be two label distributions. We have

$$\begin{aligned}
\left| \psi(\boldsymbol{y}^{(i)}; \boldsymbol{\rho}, \boldsymbol{\eta}) - \psi(\boldsymbol{y}^{(j)}; \boldsymbol{\rho}, \boldsymbol{\eta}) \right| &= \left| \min\{\hat{\rho}_i, \hat{\eta}_i\} - \min\{\hat{\rho}_j, \hat{\eta}_j\} \right| \\
&\leq \max\left\{ |\hat{\rho}_i - \hat{\rho}_j|, |\hat{\eta}_i - \hat{\eta}_j| \right\} \\
&= \max\left\{ \left| \sum_{r \in [m]} \rho_r(y_r^{(i)} - y_r^{(j)}) \right|, \left| \sum_{r \in [m]} \eta_r(y_r^{(i)} - y_r^{(j)}) \right| \right\},
\end{aligned} \tag{15}$$

where the inequality follows from the 1-Lipschitz property of the $\min$ operator w.r.t. the $\|\cdot\|_\infty$ norm. By Hölder's inequality, for any $p^{-1} + q^{-1} = 1$, we have

$$\left| \sum_{r \in [m]} \rho_r(y_r^{(i)} - y_r^{(j)}) \right| \leq \|\boldsymbol{\rho}\|_p \|\boldsymbol{y}^{(i)} - \boldsymbol{y}^{(j)}\|_q, \quad \left| \sum_{r \in [m]} \eta_r(y_r^{(i)} - y_r^{(j)}) \right| \leq \|\boldsymbol{\eta}\|_p \|\boldsymbol{y}^{(i)} - \boldsymbol{y}^{(j)}\|_q. \tag{16}$$

Combining the above inequalities yields

$$\left| \psi(\boldsymbol{y}^{(i)}; \boldsymbol{\rho}, \boldsymbol{\eta}) - \psi(\boldsymbol{y}^{(j)}; \boldsymbol{\rho}, \boldsymbol{\eta}) \right| \leq \max\{\|\boldsymbol{\rho}\|_p, \|\boldsymbol{\eta}\|_p\} \|\boldsymbol{y}^{(i)} - \boldsymbol{y}^{(j)}\|_q. \tag{17}$$

Then, we show that the above Lipschitz bound is tight. Without loss of generality, we assume that there exists a label distribution $\boldsymbol{y}^\star$ such that $\psi(\boldsymbol{y}^\star; \boldsymbol{\rho}, \boldsymbol{\eta}) = \hat{\rho}^\star \leq \hat{\eta}^\star$. Due to the definition of dual norm, there exists a vector $\boldsymbol{v}$ such that $\|\boldsymbol{v}\|_q = 1$ and $\sum_{r \in [m]} \rho_r v_r = \|\boldsymbol{\rho}\|_p$. Let $\boldsymbol{y}' = \boldsymbol{y}^\star + \varepsilon \boldsymbol{v}$ for sufficiently small $\varepsilon > 0$. Then, we have

$$\frac{|\psi(\boldsymbol{y}'; \boldsymbol{\rho}, \boldsymbol{\eta}) - \psi(\boldsymbol{y}^\star; \boldsymbol{\rho}, \boldsymbol{\eta})|}{\|\boldsymbol{y}' - \boldsymbol{y}^\star\|_q} = \frac{|\hat{\rho}' - \hat{\rho}^\star|}{\|\boldsymbol{y}' - \boldsymbol{y}^\star\|_q} = \frac{\varepsilon \left| \sum_{r \in [m]} \rho_r v_r \right|}{\varepsilon} = \|\boldsymbol{\rho}\|_p. \tag{18}$$

An analogous argument holds if we assume that there exists a label distribution $\boldsymbol{y}^\star$ such that $\psi(\boldsymbol{y}^\star; \boldsymbol{\rho}, \boldsymbol{\eta}) = \hat{\eta}^\star < \hat{\rho}^\star$.

□

### A.4. Proof of Theorem 3.2

*Proof.* **Fixed KL:** We invoke the Pinsker's inequality, which states that $\|\boldsymbol{y} - \tilde{\boldsymbol{y}}\|_1 \leq \sqrt{2\mathcal{D}_{\mathrm{KL}}(\boldsymbol{y}\|\tilde{\boldsymbol{y}})}$. By setting $p = \infty$ and $q = 1$ in Proposition 3.2, we have

$$|\psi(\boldsymbol{y}; \boldsymbol{\rho}, \boldsymbol{\eta}) - \psi(\tilde{\boldsymbol{y}}; \boldsymbol{\rho}, \boldsymbol{\eta})| \leq \max\{\|\boldsymbol{\rho}\|_\infty, \|\boldsymbol{\eta}\|_\infty\} \|\boldsymbol{y} - \tilde{\boldsymbol{y}}\|_1. \tag{19}$$

Combining the above two inequalities yields the upper bound. According to Definition 3.4, under a redistribution operation that transfers a mass $\delta > 0$ from label $\ell_i$ to label $\ell_j$, if $\rho_i = \rho_j$ and $\eta_i = \eta_j$ hold, then the divisiveness value retains unchanged, irrespective of the change in KL. Thus, the trivial lower bound $|\psi(\boldsymbol{y}; \boldsymbol{\rho}, \boldsymbol{\eta}) - \psi(\tilde{\boldsymbol{y}}; \boldsymbol{\rho}, \boldsymbol{\eta})| \geq 0$ holds.

**Fixed Euclidean Distance:** By setting $p = 2$ and $q = 2$ in Proposition 3.2, we have

$$|\psi(\boldsymbol{y}; \boldsymbol{\rho}, \boldsymbol{\eta}) - \psi(\tilde{\boldsymbol{y}}; \boldsymbol{\rho}, \boldsymbol{\eta})| \leq \max\{\|\boldsymbol{\rho}\|_2, \|\boldsymbol{\eta}\|_2\} \|\boldsymbol{y} - \tilde{\boldsymbol{y}}\|_2. \tag{20}$$

Similar to the previous case, the lower bound $|\psi(\boldsymbol{y}; \boldsymbol{\rho}, \boldsymbol{\eta}) - \psi(\tilde{\boldsymbol{y}}; \boldsymbol{\rho}, \boldsymbol{\eta})| \geq 0$ holds. □

### A.5. Experiments

The experimental results on `M²B`, `Flickr`, and `Twitter` are presented in Table 2.

*Table 2.* Experimental results on M²B, Flickr, and Twitter, formatted as "mean ± std".

| Algorithms | Cheby. ↓ | Clark ↓ | KLD ↓ | Cosine ↑ | Spear. ↑ | $\mu$ ↑ | DVSE ↓ |
|---|---|---|---|---|---|---|---|
| | | | | M²B | | | |
| RG4LDL | ● $.4496_{\pm.015}$ | ● $1.5644_{\pm.014}$ | ● $.6781_{\pm.041}$ | ● $.7148_{\pm.016}$ | ● $.6373_{\pm.030}$ | ● $36.57\%_{\pm.018}$ | ● $.0977_{\pm.007}$ |
| SA-BFGS | ● $.3826_{\pm.021}$ | ● $1.2228_{\pm.028}$ | ● $.6878_{\pm.091}$ | ● $.7231_{\pm.025}$ | $.6886_{\pm.026}$ | ● $53.06\%_{\pm.029}$ | ● $.0977_{\pm.008}$ |
| DPA | ● $.3847_{\pm.022}$ | ○ $1.1734_{\pm.030}$ | ● $.7013_{\pm.089}$ | ● $.7187_{\pm.026}$ | ○ $.6936_{\pm.024}$ | ● $52.95\%_{\pm.031}$ | ● $.0974_{\pm.008}$ |
| LCLR | ● $.3867_{\pm.022}$ | ● $1.2516_{\pm.036}$ | ● $.7303_{\pm.098}$ | ● $.7152_{\pm.026}$ | $.6862_{\pm.026}$ | ● $52.13\%_{\pm.031}$ | ● $.0971_{\pm.008}$ |
| LRR | ● $.3793_{\pm.021}$ | ○ $1.1675_{\pm.028}$ | ● $.6664_{\pm.088}$ | ● $.7271_{\pm.025}$ | ○ $.6986_{\pm.024}$ | ● $53.90\%_{\pm.029}$ | ● $.0970_{\pm.008}$ |
| LDLLC | ● $.3845_{\pm.022}$ | ○ $1.1733_{\pm.030}$ | ● $.7003_{\pm.089}$ | ● $.7189_{\pm.026}$ | ○ $.6939_{\pm.025}$ | ● $52.99\%_{\pm.031}$ | ● $.0963_{\pm.008}$ |
| DF-LDL | $.3699_{\pm.019}$ | ○ $1.1011_{\pm.025}$ | ● $.5743_{\pm.072}$ | ● $.7457_{\pm.023}$ | ○ **$.7049_{\pm.022}$** | ● $55.36\%_{\pm.027}$ | ● $.0887_{\pm.006}$ |
| $\delta$-LDL | **$.3599_{\pm.017}$** | ○ $1.0966_{\pm.023}$ | $.4882_{\pm.053}$ | **$.7768_{\pm.020}$** | ● $.6943_{\pm.023}$ | **$58.25\%_{\pm.025}$** | ● $.0872_{\pm.007}$ |
| AA-$k$NN | ● $.3721_{\pm.020}$ | ○ **$1.0566_{\pm.031}$** | ● $.6179_{\pm.074}$ | ● $.7426_{\pm.023}$ | ○ $.7022_{\pm.026}$ | ● $55.21\%_{\pm.028}$ | $.0854_{\pm.006}$ |
| LDLSF | ● $.3865_{\pm.018}$ | ● $1.5441_{\pm.031}$ | ● $2.0495_{\pm.497}$ | ● $.7363_{\pm.022}$ | ● $.6405_{\pm.026}$ | ● $32.95\%_{\pm.030}$ | $.0852_{\pm.007}$ |
| LDL-DVS | $.3642_{\pm.016}$ | $1.1871_{\pm.022}$ | **$.4839_{\pm.047}$** | **$.7768_{\pm.018}$** | $.6847_{\pm.019}$ | $57.76\%_{\pm.021}$ | **$.0845_{\pm.006}$** |
| | | | | Flickr | | | |
| RG4LDL | ● $.3591_{\pm.005}$ | ○ $2.1686_{\pm.007}$ | ● $.7491_{\pm.012}$ | ● $.7368_{\pm.006}$ | ● $.4532_{\pm.011}$ | ● $33.64\%_{\pm.007}$ | ● $.2000_{\pm.004}$ |
| LDLLC | ● $.3383_{\pm.005}$ | ○ $2.1667_{\pm.007}$ | ● $.6890_{\pm.011}$ | ● $.7611_{\pm.005}$ | ● $.4907_{\pm.010}$ | ● $38.29\%_{\pm.006}$ | ● $.1777_{\pm.003}$ |
| DPA | ● $.3377_{\pm.005}$ | ○ $2.1666_{\pm.007}$ | ● $.6871_{\pm.010}$ | ● $.7619_{\pm.005}$ | ● $.4924_{\pm.009}$ | ● $38.41\%_{\pm.006}$ | ● $.1771_{\pm.003}$ |
| LRR | ● $.3375_{\pm.005}$ | ○ $2.1667_{\pm.007}$ | ● $.6871_{\pm.010}$ | ● $.7619_{\pm.005}$ | ● $.4913_{\pm.009}$ | ● $38.42\%_{\pm.006}$ | ● $.1768_{\pm.003}$ |
| LDLSF | ● $.3355_{\pm.005}$ | ○ $2.1619_{\pm.007}$ | ● $.7493_{\pm.021}$ | ● $.7623_{\pm.004}$ | ● $.4912_{\pm.008}$ | ● $37.98\%_{\pm.006}$ | ● $.1756_{\pm.003}$ |
| SA-BFGS | ● $.3341_{\pm.005}$ | ○ $2.1675_{\pm.007}$ | ● $.6788_{\pm.010}$ | ● $.7649_{\pm.005}$ | ● $.4946_{\pm.008}$ | ● $39.11\%_{\pm.006}$ | ● $.1739_{\pm.003}$ |
| LCLR | ● $.3480_{\pm.062}$ | ○ $2.1822_{\pm.065}$ | ● $1.0011_{\pm2.201}$ | ● $.7439_{\pm.073}$ | ● $.4705_{\pm.070}$ | ● $36.67\%_{\pm.070}$ | ● $.1739_{\pm.018}$ |
| DF-LDL | ● $.3314_{\pm.005}$ | ○ $2.1694_{\pm.007}$ | ● $.6750_{\pm.010}$ | ● $.7662_{\pm.005}$ | ● $.4890_{\pm.009}$ | ● $39.45\%_{\pm.006}$ | ● $.1727_{\pm.003}$ |
| AA-$k$NN | ● $.3552_{\pm.005}$ | ○ **$2.1284_{\pm.008}$** | ● $1.8463_{\pm.096}$ | ● $.6836_{\pm.007}$ | ● $.3674_{\pm.012}$ | ● $26.84\%_{\pm.009}$ | ● $.1599_{\pm.004}$ |
| $\delta$-LDL | ● $.3057_{\pm.005}$ | ○ $2.1958_{\pm.007}$ | ● $.6401_{\pm.012}$ | ● $.7757_{\pm.006}$ | ● $.5121_{\pm.009}$ | ● $44.37\%_{\pm.008}$ | ● $.1292_{\pm.003}$ |
| LDL-DVS | **$.2934_{\pm.005}$** | $2.2026_{\pm.007}$ | **$.6127_{\pm.012}$** | **$.7867_{\pm.006}$** | **$.5258_{\pm.009}$** | **$46.50\%_{\pm.008}$** | **$.1185_{\pm.003}$** |
| | | | | Twitter | | | |
| LCLR | ● $.3881_{\pm.057}$ | ○ $2.3819_{\pm.033}$ | ● $.8324_{\pm.346}$ | ● $.7561_{\pm.098}$ | ● $.4832_{\pm.101}$ | ● $42.14\%_{\pm.099}$ | ● $.1896_{\pm.032}$ |
| RG4LDL | ● $.3644_{\pm.007}$ | ○ $2.3715_{\pm.006}$ | ● $.7386_{\pm.017}$ | ● $.8016_{\pm.008}$ | ● $.5155_{\pm.009}$ | ● $46.06\%_{\pm.012}$ | ● $.1854_{\pm.005}$ |
| LDLLC | ● $.3533_{\pm.005}$ | ○ $2.3715_{\pm.006}$ | ● $.7064_{\pm.010}$ | ● $.8081_{\pm.005}$ | ● $.5266_{\pm.009}$ | ● $48.10\%_{\pm.007}$ | ● $.1752_{\pm.004}$ |
| LRR | ● $.3532_{\pm.005}$ | ○ $2.3715_{\pm.006}$ | ● $.7065_{\pm.010}$ | ● $.8081_{\pm.005}$ | ● $.5258_{\pm.009}$ | ● $48.10\%_{\pm.008}$ | ● $.1749_{\pm.003}$ |
| DPA | ● $.3536_{\pm.005}$ | ○ $2.3715_{\pm.006}$ | ● $.7072_{\pm.010}$ | ● $.8078_{\pm.005}$ | ● $.5266_{\pm.009}$ | ● $48.05\%_{\pm.007}$ | ● $.1748_{\pm.004}$ |
| LDLSF | ● $.3518_{\pm.005}$ | ○ $2.3601_{\pm.006}$ | ● $.7651_{\pm.022}$ | ● $.8078_{\pm.005}$ | ● $.5225_{\pm.008}$ | ● $47.67\%_{\pm.007}$ | ● $.1723_{\pm.003}$ |
| SA-BFGS | ● $.3516_{\pm.005}$ | ○ $2.3725_{\pm.006}$ | ● $.7038_{\pm.010}$ | ● $.8079_{\pm.005}$ | ● $.5249_{\pm.008}$ | ● $48.32\%_{\pm.007}$ | ● $.1717_{\pm.003}$ |
| DF-LDL | ● $.3530_{\pm.005}$ | ○ $2.3740_{\pm.006}$ | ● $.7096_{\pm.010}$ | ● $.8059_{\pm.005}$ | ● $.5159_{\pm.008}$ | ● $47.91\%_{\pm.007}$ | ● $.1692_{\pm.003}$ |
| AA-$k$NN | ● $.3777_{\pm.005}$ | ○ **$2.1347_{\pm.012}$** | ● $2.6094_{\pm.139}$ | ● $.7268_{\pm.007}$ | ● $.4386_{\pm.010}$ | ● $33.87\%_{\pm.011}$ | ● $.1522_{\pm.003}$ |
| $\delta$-LDL | ● $.2998_{\pm.005}$ | ○ $2.4044_{\pm.005}$ | ● $.6375_{\pm.013}$ | ● $.8219_{\pm.006}$ | ● $.5447_{\pm.009}$ | ● $54.70\%_{\pm.009}$ | ● $.1216_{\pm.003}$ |
| LDL-DVS | **$.2948_{\pm.005}$** | $2.4056_{\pm.006}$ | **$.6238_{\pm.013}$** | **$.8259_{\pm.006}$** | **$.5512_{\pm.009}$** | **$55.56\%_{\pm.009}$** | **$.1131_{\pm.004}$** |

