# OpenReview forum: "Divisiveness-Consistent Label Distribution Learning"
_ICML.cc/2026/Conference — ICML 2026 regular_

### Official Review · Reviewer_9P7t · 2026-03-04

**Soundness:** 3
**Presentation:** 3
**Significance:** 4
**Originality:** 3
**Overall Recommendation:** 5
**Confidence:** 4

**Summary:**

This paper introduces a decision-relevant characteristic in label distribution learning: divisiveness, which refers to the degree of polarization between semantically opposing labels, and investigate the “divisiveness” problem in LDL. This paper reveals the inconsistencies between conventional loss functions (e.g., KL divergence, mean squared error) and divisiveness error: Multiple predictions close to the true label distribution under conventional loss functions may exhibit significantly different degree of divisiveness. To address this issue, the paper makes three contributions: (1) This paper proposes a formal definition of divisiveness metrics satisfying the polarity monotonicity axiom; (2) this paper theoretically explains why conventional loss functions fail to adequately capture divisiveness; (3) this paper designs a pairwise divisiveness loss function as an unbiased proxy for the original divergence error. The proposed method is evaluated across multiple benchmark datasets, with experimental results demonstrating the improvements over both the conventional LDL metrics and the divisiveness error.

**Compliance With Llm Reviewing Policy:**

Affirmed.

**Final Justification:**

Thanks for the response. The rebuttal resovled my major concerns and reinforced my prior assessment.

**Key Questions For Authors:**

1. The pairwise loss $\hat{\mathcal{L}}_\psi$ involves summing over $m(m-1)$ label pairs. For datasets with a large number of labels, how can we address the issue of excessive computational overhead?

2. How much bias does the softmin function in equation (7) introduce when approximating $\min{\cdot,\cdot}$, and how does the choice of $k$ affect the accuracy of the approximation?

**Limitations:**

Yes

**Strengths And Weaknesses:**

Strengths:
1. This paper demonstrates strong technical soundness in multiple aspects. The authors propose an axiomatic property that any divisiveness metric should satisfy and prove that the proposed divisiveness metric satisfies this property in Theorem 3.1.

2. Furthermore, the authors extensively analyze why conventional LDL metrics cannot serve as the divisiveness error metric in Proposition 3.1. The Lipschitz continuity of the proposed divisiveness metric is proven in Proposition 3.2, and a bound relationship between divisiveness error and conventional loss functions is derived in Theorem 3.2 to theoretically reveal the inconsistencies between conventional loss functions and divisiveness error.

3. The experimental employs ten-fold cross-validation repeated ten times, statistical significance testing, and comparisons against ten state-of-the-art LDL methods. This paper is well-structured and clearly written. Figures 1 and 2 effectively illustrate the research motivation. Figure 3 clearly articulates the design rationale for divisiveness metric. Definitions and theorems are precisely presented.

4. This paper identifies a previously overlooked characteristic in label distributions and provides axiomatic metrics and learning methods to address this issue, which establishes a fundamental principle for subsequent research on divisiveness of label distribution and advances the practical application of label distribution learning methods.


Weaknesses:

1. $\hat{\mathcal{L}}_\psi$ involves summing $O(m^2)$ terms. For applications with large numbers of labels, this computational overhead may become a bottleneck. This paper should discuss the strategies for scaling to high-dimensional label spaces.

2.  Equation (7) uses a softmin function with hyperparameter $k$ to approximate $\min{\cdot,\cdot}$ operation for global differentiability. However, this paper has not analyzed the magnitude of bias introduced by this approximation or how the choice of $k$ affects the accuracy of the approximate estimates. When $k$ is finite, the approximate gradient may exhibit bias relative to the true gradient, which can impact the model performance.

---

> ### Author Rebuttal · Authors · 2026-03-28
>
> Dear Reviewer 9P7t,
>
> Thank you for your insightful and constructive comments on our paper. Below, we provide a point-by-point response to the issues you have raised.
>
> ### Q1 (Computational overhead of the pairwise term in large label spaces)
> 1. In typical LDL scenarios encountered in practice—and consistent with all benchmark datasets used in this study—the number of labels usually ranges between 5 and 20. Within this scale, the computational overhead introduced by the pairwise term is negligible relative to the overall model cost.
> 2. To address the computational overhead problem in future implementations, we preliminarily propose two practical engineering strategies:
> - Constructing pairwise terms only over label subsets where semantic polarity is non-zero, thereby sparsifying the interaction graph.
> - Using random sampling of label pairs within each mini-batch to approximate the pairwise regularization term, which can significantly reduce per-iteration complexity while preserving learning stability.
> ### Q2 (Impact of the parameter $k$ on the approximation error of softmin)
> The absolute deviation between the min function and the softmin function adopted in our paper can be formulated as:
> $$
> \Bigg\vert \frac{a e^{-ka} + b e^{-kb}}{e^{-ka} + e^{-kb}} - \min(a, b)\Bigg\vert = \frac{\Delta}{e^{k\Delta}+1},
> $$
> where $\Delta =\vert b-a \vert$, and $k$ is the smoothness parameter.
>
> According to the above equations, the relationship between $k$ and the approximation error can be summarized as follows:
>
> - Monotonicity: Strictly monotonic decreasing with respect to $k$.
> - Upper Bound: As $k\rightarrow 0^+$, the approximation error $\rightarrow \frac{\Delta}{2}$.
> - Lower Bound: As $k\rightarrow +\infty$, the approximation error $\rightarrow 0$.
> - Decay Rate: The approximation error decays exponentially as $\mathcal{O}(e^{-k\Delta})$.
>
> We hope these clarifications address your concerns. We are grateful for your insightful feedback, which has helped us improve the paper.
>
> Best regards.

---

> > ### Author Rebuttal · Reviewer_9P7t · 2026-04-01
> >
> > I have carefully read the rebuttal. The response have sovled my major concerns. According to the contribution of the proposed method. I still intend to give the positive score.

---

> > > ### Author Response · Authors · 2026-04-03
> > >
> > > Dear Reviewer 9P7t,
> > >
> > > Thank you for your thoughtful review. It is your insightful comments that have made our work more complete.
> > >
> > > Best regards.

---

### Official Review · Reviewer_xfPU · 2026-03-09

**Soundness:** 3
**Presentation:** 3
**Significance:** 3
**Originality:** 3
**Overall Recommendation:** 4
**Confidence:** 5

**Summary:**

This paper proposes a novel Label Distribution Learning (LDL) algorithm named LDL-DVS, which focuses on the preservation of divisiveness among semantically opposing labels. LDL-DVS introduces a principled divisiveness measure based on polarity vectors and applies a pairwise divisiveness loss to provide an unbiased estimation for the divisiveness-consistent error. Extensive experiments show that LDL-DVS performs well on several real-world LDL datasets across both the newly proposed Divisiveness-Consistent Error and traditional LDL metrics. The theoretical analysis verifies the inconsistency between conventional loss functions and divisiveness error.

**Compliance With Llm Reviewing Policy:**

Affirmed.

**Key Questions For Authors:**

1. The concept of 'divisiveness' in this paper relies on the existence of semantically opposing or polar labels. However, in some LDL scenarios, the labels may not exhibit clear semantic conflicts. For example, in image datasets, labels represent the proportions of different objects, and they seems not conflict with each other. Could the authors clarify whether the proposed LDL-DVS algorithm is applicable to such datasets? It would be helpful to discuss the generalizability of the proposed algorithm.

2. As is said in this paper, conventional loss functions (such as KL divergence and MSE loss) are inconsist with the divisiveness error. However, in the overall loss function, the KL divergence and the proposed surrogate divisiveness loss are optimized jointly. As they conflict with each other, is it still reasonable to put them together?

**Limitations:**

yes

**Strengths And Weaknesses:**

**Strengths:**
1. The problem studied in this paper is novel and interesting in the label distribution learning area, indicating the necessity of taking label divisiveness into consideration.
2. The paper presents a reasonable approach to capture divisiveness information in label distribution learning problems.
3. The paper is well-supported by both rich theoretical analysis and extensive experimental evaluations.

**Weaknesses:**
1. Out of the ten comparative algorithms, only two are from 2025 (RG4LDL and $\delta$-LDL) and others are relatively outdated. More recent state-of-the-art LDL algorithms should be incorporated to provide a more convincing demonstration of the proposed method's performance.

---

> ### Author Rebuttal · Authors · 2026-03-28
>
> Dear Reviewer xfPU,
>
> Thank you for your thoughtful and constructive comments on our manuscript. We have carefully considered each point and provided our responses below.
>
> **Regarding the weakness on baseline algorithms:**
>
> Due to space constraints, we originally selected representative algorithms, including correlation modeling, noise robustness, and ranking-oriented approaches. Our baseline setting already includes two latest methods, RG4LDL (Tan et al., 2025) and $\delta$-LDL (Li et al., 2025), as well as a number of representative LDL algorithms developed in recent years, such as low-rank correlation modeling (LCLR, LDLLC), ranking consistency methods (LRR, DPA), and decomposition-fusion frameworks (DF-LDL). Nevertheless, we acknowledge your concern regarding the inclusion of additional recent state-of-the-art methods. Therefore, we have conducted further experiments to compare our method with $\mathcal{S}$-LRR (Haitao Wu, Weiwei Li, Xiuyi Jia. Divide and conquer: Learning label distributions with subtasks. ICML 2025, 67408-67426) and R$k$NN-LDL (Jing Wang, Fu Feng, Jianhui Lv, Xin Geng. Residual k-Nearest Neighbors Label Distribution Learning. Pattern Recognition, 2025, 158:111006) on RAF-ML dataset. The results are presented in the table below.
> | Method | Cheby. ($\downarrow$) | Clark ($\downarrow$) | KLD ($\downarrow$) | Cosine ($\uparrow$) | Spear. ($\uparrow$) | $\mu\ (\uparrow)$ | DVSE ($\downarrow$) |
> | :--- | :--- | :--- | :--- | :--- | :--- | :--- | :--- |
> | $\mathcal{S}$-LRR | .2232$ _{±.007} $ | 1.6006$ _{±.018} $ | .3483$ _{±.019} $ | .8893$ _{±.009} $ | .7476$ _{±.018} $ | 60.29%$ _{±.016} $ | .1222$ _{±.005} $ |
> | R$k$NN-LDL | .3590$ _{±.035} $ | 1.5501$ _{±.024} $ | .6165$ _{±.068} $ | .6705$ _{±.061} $ | .4195$ _{±.033} $ | 27.04%$ _{±.045} $ | .1235$ _{±.017} $ |
> | LDL-DVS | **.1493$ _{±.005} $** | **1.4210$ _{±.016} $** | **.1873$ _{±.011} $** | **.9300$ _{±.005} $** | **.7863$ _{±.009} $** | **76.67%$ _{±.010} $** | **.0764$ _{±.003} $** |
>
> **Regarding Question 1:**
>
> The applicability of the proposed divisiveness measure presupposes that the label space contains semantically opposing label pairs. In such scenarios, decision-makers are indeed concerned with whether the opinions (labels) are polarized, making the divisiveness measure a meaningful risk indicator. For tasks that lack an explicit structure of mutually conflicting semantics, the polarity vector can be treated as a zero vector, implying that the degree of divisiveness is also zero. In these tasks, decision-makers pay no attention to divisiveness risk (since it is always zero), and therefore have no need for the algorithm we propose. Therefore, we do not recommend applying the proposed method to such tasks. In the revised version, we will clarify the applicability of LDL-DVS by more explicitly delineating its scope, thereby better articulating the generalizability of the algorithm.
>
> **Regarding Question 2:**
>
> We acknowledge your concern regarding the joint optimization of two inconsistent objectives and offer the following clarifications:
> 1. By "inconsistency," we refer to the fact that optimizing one objective does not guarantee the optimality of the other one, rather than implying that the two loss functions cannot be jointly optimized. Theorem 3.2 and Figure 2 illustrate the limitations that optimizing KL/MSE alone does not ensure a small DVSE. This highlights precisely why the two terms should be optimized together; if optimizing one naturally led to optimality of the other, there would be no need to consider both.
>
> 2. From an optimization perspective, KL divergence and DVSE are complementary in terms of their local gradients. KL ensures that the predicted label distribution does not deviate excessively from the ground-truth, while DVSE further encourages accurate prediction of the divisiveness associated with the ground-truth. Without the KL term, optimizing DVSE alone would yield numerous solutions with identical DVSE values but vastly different distributional shapes. Conversely, optimizing only KL divergence leads to the phenomenon illustrated in Figure 2, where a U-shaped distribution is distorted into a bell-shaped one. Combining the two terms via a linear weighting essentially integrates the constraints from two complementary perspectives.
>
> We hope these clarifications adequately address your concerns. Thank you again for your valuable feedback, which has helped us improve the clarity and rigor of our work.
>
> Sincerely.

---

> > ### Author Rebuttal · Reviewer_xfPU · 2026-04-02
> >
> > My previous concerns have been fully addressed in this rebuttal. However, as the discussion regarding **Question 1** confirmed that the proposed algorithm indeed suffers from limited applicable scenarios, I will keep my original score.

---

> > > ### Author Response · Authors · 2026-04-03
> > >
> > > Dear Reviewer xfPU,
> > >
> > > Thank you for acknowledging that our responses have addressed your previous concerns.
> > >
> > > You are right that the proposed algorithm is not intended for all tasks. However, we would like to respectfully clarify that universal applicability across all possible tasks is not the objective of our work. Instead, we explicitly delineate the scope of our method so that practitioners can clearly understand when it should and should not be used. Within its intended problem domain, our algorithm provides an effective and reliable solution. Therefore, a more appropriate way to articulate the "applicability" is that our algorithm is tailored for a clearly-defined and important class of real-world tasks that involve the risk of divisiveness arising from decision-making.
> > >
> > > Thank you again for your thoughtful review.
> > >
> > > Sincerely.

---

### Official Review · Reviewer_X5xm · 2026-03-12

**Soundness:** 3
**Presentation:** 3
**Significance:** 2
**Originality:** 3
**Overall Recommendation:** 3
**Confidence:** 2

**Summary:**

This paper studies label distribution learning from the perspective of divisiveness, namely the extent to which a label distribution reflects dissension between semantically opposing labels. The authors argue that standard LDL losses such as KL divergence and MSE focus on global distributional similarity and can therefore miss this practically relevant property. To address this, they define a divisiveness measure using polarity vectors, show that conventional LDL objectives are not well aligned with divisiveness preservation, and introduce an additional pairwise surrogate loss to encourage divisiveness-consistent predictions.

**Compliance With Llm Reviewing Policy:**

Affirmed.

**Final Justification:**

The rebuttal adequately addressed most of the concrete concerns I raised. Moreover, the authors’ follow-up response clarifying the LDL setting was helpful and indeed improved my understanding of the paper.

However, I still have a more fundamental reservation about the problem formulation itself: while the proposed notion of *divisiveness* is interesting and potentially useful, it also feels somewhat task-specific and dependent on hand-crafted semantic structure. That said, this impression may partly stem from my limited familiarity with the area, as reflected in my confidence rating of 2. Therefore, I prefer to keep my original score rather than revise it upward based solely on the rebuttal.

Overall, I view the paper as having real merits, but I remain insufficiently convinced about the generality of the setting to change my recommendation. Therefore, I maintain my score at 3 (Weak Reject) while increasing the subratings for soundness, presentation, and originality.

**Key Questions For Authors:**

Q1. How sensitive are the results to the manually specified polarity vectors and the choice of which labels are excluded as neutral

Q2. How would the method behave on LDL tasks with many labels or with no clear binary semantic opposition between labels

**Limitations:**

See weaknesses and questions.

**Strengths And Weaknesses:**

S1. The motivation is intuitive and well presented. The paper clearly explains why matching the overall label distribution is not necessarily the same as preserving polarization-like structure, and the examples in Figures 1 and 2 make this point effectively.

S2. The method is technically coherent. The paper does not merely introduce a new metric, but also analyzes why directly optimizing divisiveness can be unstable and proposes a surrogate objective with supporting theory and ablation results.

S3. The paper is overall well organized, and the theoretical motivation is closely connected to the algorithmic design.

W1. The notion of divisiveness is somewhat hand-crafted and task-specific. It requires predefined polarity vectors or manual grouping of labels into positive and negative sets, which may be ambiguous or unnatural for many LDL problems. More importantly, reducing divisiveness to total positive-versus-negative mass may ignore richer multi-modal or multi-faction structure within the label space.

W2. The method introduces a real trade-off with standard LDL objectives. The paper itself notes that overemphasizing divisiveness can hurt conventional metrics, Clark distance is often not competitive, and the method does not achieve the best divisiveness result on the Painting dataset.

W3. There are also some stability and scalability concerns. The pairwise construction becomes heavier as the label space grows, and Figure 4(c) shows that larger smoothing values increase training time, with very large values even causing numerical failure.

---

> ### Author Rebuttal · Authors · 2026-03-28
>
> Dear Reviewer X5xm,
>
> Thank you very much for your constructive comments on our work. We have carefully considered each point and have prepared our responses as follows.
>
> **Response to W1:**
>
> First, we acknowledge that divisiveness is task-specific and hand-crafted. But, this does not diminish our contributions in addressing LDL decision risk. To clarify this, we draw an analogy to cost-sensitive learning (CSL). In CSL, the misclassification cost is also typically task-specific and hand-crafted, relying on domain knowledge. Yet, this does not undermine the merits of CSL over accuracy-oriented classifiers in managing decision risks. Similarly, compared to KL/MSE-oriented LDL methods, our method shows clear advantages in addressing the divisiveness in LDL decision.
>
> Second, we proposed several methods for constructing polarity vector. In our paper, we already present two methods:
> - For ordinal labels, we employ a "staircase" method to define the degree of positivity/negativity.
> - For discrete emotional labels, we can leverage established psychological theories to assign positivity and negativity.
>
> Beyond these, two additional promising methods are:
> - Domain knowledge or downstream task definitions.
> - Data-driven approach, such as projecting the labels onto a one-dimensional space.
>
> **Regarding the multi-faction:**
>
> We agree that the label space can be multi-faction sometimes, where divisiveness arises from multiple competing groups rather than a binary opposition. But, we argue that this does not impact the contributions of our work, as multi-faction divisiveness is a generalization built upon the binary-faction foundation. Our proposal can be directly extended to the multi-faction scenarios with minor revisions. This is analogous to the relationship between single-label and multi-label learning: the latter builds upon the former, and the limitations of single-label methods in multi-label cases do not negate their significance.
>
> **Response to W2:**
>
> We agree with the trade-off between KL objectives and the divisiveness loss. We would like to clarify that this trade-off is an inherent property of the divisiveness-consistent LDL problem, not a shortcoming of our method. In fact, the trade-off prevents existing methods from addressing divisiveness problem, serving as the central motivation for our methodology.
> We acknowledge that our method effectively mitigates, but does not completely eliminate, the trade-off problem. Nevertheless, our empirical results show that the trade-off is ignorable. Across most datasets, our LDL-DVS consistently achieves the best or near-best performance in terms of DVSE and KLD. While some degradation is observed in Clark, this is attributable to Clark’s sensitivity to sparse label distributions.
>
> **Response to W3:**
> 1. *Complexity of pairwise construction:* Due the character limit, please refer to our response to Q1 of Reviewer 9P7t.
>
> 2. *Smoothing parameter $k$ and numerical stability:* Figure 4(c) shows that large $k$ may lead to numerical instability and increased training time, serving as a practical guide for parameter tuning. In all reported results, we uniformly set $k=10$, which is stable and efficient.
>
> **Response to Q1:**
>
> We have conducted further experiments and will add the following analyses to the revised version within the page limits:
> 1. *Sensitivity to perturbations in polarity intensity:* We randomly perturb the staircase polarity vector on the M2B training dataset (test set is unchanged), i.e., $\rho=[0,0,0.1,0.4,1]; \eta=[1,0.4,0.1,0,0]$. Results (Cheby.: 0.3656; Clark: 1.1928; KLD: 0.4838; Cosine: 0.7767; Spear.:0.6825; $\mu$: 57.27%; DVSE: 0.0872) show a minimal changes in performance.
>
> 2. *Sensitivity to the neutral labels:* On the Music dataset, we move some weak emotion labels from neutral to either positive or negative, i.e., $\rho=[1,0,1,0,1,1,1,0.5,0.5],\eta=[0,1,0,1,0,0,0,0.5,0.5]$. Results (Cheby.: 0.0735; Clark: 0.7143; KLD: 0.1052; Cosine: 0.9224; Spear.:0.4958; $\mu$: 39.84%; DVSE: 0.0477) show a minimal changes in performance. We will include these findings in the final version, space permitting.
>
> **Response to Q2:**
> 1. *When no opposing labels exist:* If a task completely lacks opposing labels, then divisiveness is not worth considering. Our method presupposes the existence of at least one interpretable "positive-negative" pair within the label space. If an LDL task involves no opposing labels, i.e., experts have no concerns about divisiveness risks, then there is no need to adopt our method. Overall, this is not a failure of the algorithm but rather an indication the scope of our proposal.
> 2. *Handling numerous labels with underlying opposition:* Even with numerous labels, high-level semantic oppositions can often be readily identified.
>
> Once again, we sincerely thank you for the valuable feedback, which has significantly helped us improve our work. If you have any further questions, please don’t hesitate to ask.
>
> Best regards.

---

> > ### Author Rebuttal · Reviewer_X5xm · 2026-04-01
> >
> > Overall, the previously raised weaknesses and questions appear to have been adequately addressed. However, due to my limited familiarity with this research area, I will maintain my original score.

---

> > > ### Author Response · Authors · 2026-04-01
> > >
> > > Dear Reviewer X5xm,
> > >
> > > Thank you for acknowledging that our responses have adequately addressed your concerns. Given that you mentioned being less familiar with this research area, we will explain our work in an accessible manner.
> > >
> > > ### What is Label Distribution Learning (LDL)
> > >
> > > To understand our work, it helps to first understand the field we are working in: **Label Distribution Learning (LDL)**.
> > > In traditional machine learning, each data point is assigned a single label or a set of multiple labels. However, in many real-world scenarios, a data point is better described by a *distribution* over multiple labels, i.e., label distribution. For example:
> > >
> > > - A facial expression might be 70% happiness, 20% surprise, and 10% sadness.
> > > - A movie might have 40% 5-star ratings, 10% 4-star, 10% 3-star, 10% 2-star, and 30% 1-star ratings.
> > >
> > > LDL is a machine learning paradigm designed to handle exactly this type of data. Instead of predicting a single class, LDL predicts a *distribution* across all possible labels, capturing the richness of the underlying phenomenon.
> > > LDL has been successfully applied in areas such as facial expression recognition, emotion analysis, movie rating prediction, and public opinion analysis. The field has developed many methods over the past decade, with most approaches focusing on making the predicted label distribution as close as possible to the true label distribution using standard distance measures like KL divergence or mean squared error (MSE).
> > >
> > > ### The Problem We Address
> > >
> > > While existing LDL methods are effective at matching the *overall shape* of the label distribution, they overlook an essential decision property: **divisiveness**.
> > > Divisiveness within label distribution refers to the propensity of a label distribution to exhibit dissension between semantically opposing labels, which is an essential indicator of the practical decision risk. Consider two movie rating distributions:
> > >
> > > - **U-shaped distribution**: Many 5-star and many 1-star ratings, few in the middle. This is highly divisive: most people either love it or hate it.
> > > - **Bell-shaped distribution**: Most ratings clustered around 3 stars. This reflects consensus: people generally agree on a moderate opinion.
> > >
> > > These two label distributions convey very different information for decision-making. A U-shaped distribution signals risk and opportunity, while a bell-shaped distribution signals safety and mediocrity. However, a model trained with standard LDL loss functions like KL divergence and MSE may fail to preserve whether a prediction is truly divisive or not.
> > >
> > > ### Our Contribution
> > >
> > > We propose a new LDL framework that explicitly accounts for divisiveness. Our work consists of three main contributions:
> > >
> > > 1. **A principled way to measure divisiveness.**
> > >    We define a mathematical measure that quantifies how divisive a label distribution is. We also introduce a theoretical property (*polarity monotonicity*) that any reasonable divisiveness measure should satisfy, and prove that our measure meets this property. This provides a solid foundation for our approach.
> > >
> > > 2. **A theoretical analysis of why existing LDL methods fail to preserve divisiveness.**
> > >    We show mathematically that conventional LDL loss functions do not guarantee the preservation of divisiveness. Even when the overall prediction error is small, the divisiveness error can still be large. This explains why existing methods are not suitable for tasks where divisiveness matters.
> > >
> > > 3. **A new loss function for LDL that preserves divisiveness.**
> > >    We design a dedicated loss term that explicitly minimizes the error in divisiveness. We also address a technical challenge: directly optimizing divisiveness leads to unstable training. To solve this, we propose a *pairwise surrogate loss* that is unbiased and ensures stable convergence. This loss is combined with standard KL divergence to maintain both overall distribution accuracy and divisiveness fidelity.
> > >
> > > Overall, our work introduces a new perspective to LDL: divisiveness, which is important for the real-world applications where polarization matters, such as predicting public opinion, assessing social risks, or analyzing consumer preferences.
> > > By providing a principled measure and an effective learning strategy, we expand the capabilities of LDL to handle tasks where the divisiveness of the distribution—not just its numerical proximity—carries critical information.
> > >
> > > We hope this explanation clearly clarifies the research field and our work.
> > >
> > > Sincerely.

---

### Decision · Program_Chairs · 2026-04-30

**Decision:**

Accept (regular)

**Comment:**

The paper introduces a theoretically sound framework for "divisiveness" in Label Distribution Learning, supported by an axiomatic definition of polarity monotonicity and a dedicated pairwise surrogate loss. Its primary weakness is a reliance on manually defined polarity vectors, which restricts the method's generalizability to tasks with explicitly known semantic oppositions. Scalability and stability also remain concerns, as the computational complexity of the pairwise construction and the potential for numerical failure with high smoothing parameters limit the framework's use in high-dimensional settings. Despite these constraints on universality, the technical depth and novel treatment of decision risk make the work a specialized contribution.